EMBO
reports

# TBK1-mediated phosphorylation of LC3C and GABARAP-L2 controls autophagosome shedding by ATG4 protease

Lina Herhaus[1], Ramachandra M Bhaskara[2,‡], Alf Håkon Lystad[3,‡], Uxía Gestal-Mato[1], Adriana Covarrubias-Pinto[1], Florian Bonn[1,†], Anne Simonsen[3] (ID), Gerhard Hummer[2,4] (ID) & Ivan Dikic[1,5,*] (ID)

## Abstract

Autophagy is a highly conserved catabolic process through which defective or otherwise harmful cellular components are targeted for degradation via the lysosomal route. Regulatory pathways, involving post-translational modifications such as phosphorylation, play a critical role in controlling this tightly orchestrated process. Here, we demonstrate that TBK1 regulates autophagy by phosphorylating autophagy modifiers LC3C and GABARAP-L2 on surface-exposed serine residues (LC3C S93 and S96; GABARAP-L2 S87 and S88). This phosphorylation event impedes their binding to the processing enzyme ATG4 by destabilizing the complex. Phosphorylated LC3C/GABARAP-L2 cannot be removed from liposomes by ATG4 and are thus protected from ATG4-mediated premature removal from nascent autophagosomes. This ensures a steady coat of lipidated LC3C/GABARAP-L2 throughout the early steps in autophagosome formation and aids in maintaining a unidirectional flow of the autophagosome to the lysosome. Taken together, we present a new regulatory mechanism of autophagy, which influences the conjugation and de-conjugation of LC3C and GABARAP-L2 to autophagosomes by TBK1-mediated phosphorylation.

Keywords ATG4; ATG8; autophagy; phosphorylation; TBK1
Subject Categories Autophagy & Cell Death; Post-translational Modifications & Proteolysis

## Introduction

The recycling of redundant cytosolic components and damaged organelles is termed autophagy. It is a highly conserved process, which increases during starvation conditions or during other cellular stresses [1,2]. Autophagy involves the formation of a phagophore, a double-membrane cup-shaped structure, which expands to enwrap and enclose the designated cellular cargo to form the autophagosome, which then fuses with lysosomes to enable enzymatic degradation of its cargo along with its inner membrane [3]. Upon induction of autophagy, small ubiquitin-like LC3 proteins (autophagy modifiers) are conjugated to phosphatidylethanolamine (PE) anchoring them to the growing phagophore membrane (LC3-II). This conjugation is carried out by the lipidation cascade enzymes (ATG3, ATG5, ATG7, ATG12, and ATG16L1) and allows cargo selection and autophagosome formation [4,5]. In humans, there are six autophagy-modifier proteins, grouped into two subfamilies: (i) LC3A, LC3B, and LC3C, and (ii) GABARAP, GABARAP-L1, and GABARAP-L2/GATE-16 [6]. Both subgroups play different roles during autophagy. The LC3 subgroup mainly mediates the elongation of the phagophore membrane, whereas the GABARAP subfamily acts at later stages during autophagosome sealing and autophagosome–lysosome fusion [7,8]. LC3 proteins undergo two processing steps, (i) an initial proteolytic cleavage of the peptide bond responsible for the conversion of pro-LC3 to active LC3 (LC3-I) and (ii) the subsequent cleavage of the amide bond for de-lipidation of LC3-II from autophagosomes to regenerate a free cytosolic LC3 pool. Both processing steps are catalyzed by ATG4 [9]. Among the four mammalian paralogs of ATG4, ATG4B is the most active protease followed by ATG4A and ATG4C/D, which exhibit minimal protease activity [10,11]. LC3s are bound by the ATG4B enzyme body and through LC3-interacting regions (LIRs) located at the

1 Institute of Biochemistry II, School of Medicine, Goethe University, Frankfurt am Main, Germany
2 Department of Theoretical Biophysics, Max Planck Institute of Biophysics, Frankfurt am Main, Germany
3 Department of Molecular Medicine, Faculty of Medicine, Institute of Basic Medical Sciences and Centre for Cancer Cell Reprogramming, Institute of Clinical Medicine, University of Oslo, Oslo, Norway
4 Institute for Biophysics, Goethe University, Frankfurt am Main, Germany
5 Buchmann Institute for Molecular Life Sciences, Riedberg Campus, Goethe University Frankfurt, Frankfurt am Main, Germany
*Corresponding author. Tel: +496963015964; Fax: +496963015577; E-mail: dikic@biochem2.uni-frankfurt.de
‡ These authors contributed equally to the work
† Present address: Immundiagnostik AG, Bensheim, Germany

N- and C-terminal flexible tails of ATG4B [12]. ATG4s are cysteine proteases that cleave peptide bonds of pro-LC3 to expose the C-terminal glycine and allow conjugation with PE. ATG4 also de-conjugates LC3-II from the outer membrane of autophagosomes preceding, or just after autophagosome–lysosome fusion by cleaving the amide bond between PE and the C-terminal glycine residue of LC3s [13]. Phosphorylation of ATG4B at S383/392 increases its protease activity [14], especially during the LC3 de-lipidation phase, whereas ULK1-mediated phosphorylation of ATG4B at S316 (in humans) [15] or at S307 (in yeast) [16] reduces pro-LC3 binding and C-terminal tail cleavage. Likewise, oxidation of ATG4 by $H_2O_2$ attenuates its activity and blocks LC3 de-lipidation [17].

The serine–threonine kinase TBK1 has been implicated in the selective degradation of depolarized mitochondria (mitophagy) and intracellular pathogens (xenophagy) [18–20]. Specific autophagic cargo marked with an ubiquitin signal is recognized by autophagy receptor proteins such as optineurin (OPTN), p62, or NDP52 [5,21]. They physically bridge the cargo to the nascent phagophore by binding to ubiquitin via their UBD and to LC3 family proteins via their conserved LIR motifs, respectively. These autophagy receptors also recruit TBK1 to the site of autophagosome formation. Local accumulation of active TBK1 phosphorylates p62 and OPTN, which increases their binding affinity for polyubiquitin chains and the LC3 family proteins, thereby driving autophagy [19,20,22–27].

In this study, we expand the role of TBK1 in the autophagic process by demonstrating its ability to directly phosphorylate LC3C on S93/96 and GABARAP-L2 on S87/88. We study the consequences of LC3 phosphorylation during autophagy and show that this phosphorylation primarily impedes ATG4-mediated processing of LC3 on the liposomes, adding a new layer of regulation.

# Results

## TBK1 phosphorylates LC3C and GABARAP-L2 *in vitro*

The serine–threonine kinase TBK1 has previously been shown to phosphorylate autophagy receptors such as OPTN and p62 [19,20,26]. To test whether recombinant TBK1 can also directly phosphorylate autophagy modifiers, we performed an *in vitro* kinase assay. LC3A, LC3C, GABARAP-L1, and GABARAP-L2 are directly phosphorylated by TBK1 *in vitro,* and the phosphorylation sites of LC3 family proteins were identified by mass spectrometry (Fig 1A

and B). Since LC3A is only weakly phosphorylated and the detected phosphorylation site of GABARAP-L1 is a tyrosine residue (most likely an *in vitro* assay artifact), we decided to further investigate the phosphorylation sites of LC3C at S93 and S96 and GABARAP-L2 at either S87 and S88, which could not be unambiguously assigned. The TBK1-mediated phosphorylation sites of LC3C (Fig EV1A) and GABARAP-L2 (Fig EV1B) are topologically equivalent and are present in surface-exposed loops (depicted in red). The position of the loop is in between β-strand 3 and α-helix 3 in both LC3C and GABARAP-L2, on the opposite face of the LIR binding pocket. This indicates that LIR-mediated interactions of LC3C might not be affected directly upon phosphorylation. The TBK1-mediated phosphorylation sites of LC3C (Fig 1C) and GABARAP-L2 (Fig 1D), marked with red triangles, are mostly conserved in orthologs from higher eukaryotes and fit to the general TBK1 consensus phosphorylation motif (Fig 1E), which prefers a hydrophobic residue (mostly leucine) after the target phosphoserine site [28]. Moreover, these phosphorylation sites are situated in solvent-accessible regions of LC3C and GABARAP-L2 (Fig 1F).

## TBK1 phosphorylates and binds LC3C and GABARAP-L2 *in cells*

To test whether TBK1 also phosphorylates LC3C in cells, HEK293T cells were SILAC-labeled and either WT TBK1 (heavy label) or TBK1 kinase dead (K38A; light label) was overexpressed along with GFP-LC3C or with GFP-GABARAP-L2. GFP proteins were immunoprecipitated and analyzed by mass spectrometry. Phosphorylation at positions S93 and S96 of LC3C was enhanced in the presence of WT TBK1 (factors 10 and 6, respectively; Fig 2A), as compared to TBK1 K38A. Similarly, the presence of WT TBK1 resulted in enhanced phosphorylation of GABARAP-L2 at S87 and S88 (by factors 13 and 2, respectively; Fig 2A). To confirm this phosphorylation event on an endogenous level, we treated cells with control or TBK1 siRNA and induced TBK1 kinase activity with CCCP treatment (Fig 2B). siRNA-mediated depletion of TBK1 in cells resulted in a twofold reduction of S96 LC3C and S87 GABARAP-L2 phosphorylation. Unfortunately, our efforts to generate phospho-specific antibodies against GABARAP-L2 S87-$PO_4$ and GABARAP-L2 S87/88-$PO_4$ failed (Fig EV1C).

Next, we visualized the phosphorylation event by using Phos-tag™ polyacrylamide gels, where phosphorylated proteins are retained by the Phos-tag reagent and appear at a higher molecular weight. Overexpression of WT TBK1, but not TBK1 kinase dead,

**Figure 1. TBK1 phosphorylates LC3C and GABARAP-L2 *in vitro*.**

A    Coomassie stain and autoradiography of SDS–PAGE after an *in vitro* kinase assay with GST-TBK1 and His-LC3 family proteins as substrates. TBK1 phosphorylates LC3A, LC3C, GABARAP-L1, and GABARAP-L2 *in vitro*.

B    Identification of phosphosites by mass spectrometry following an *in vitro* TBK1 kinase assay.

C, D  Alignments showing selected orthologs of human LC3C (C) and human GABARAP-L2 (D) highlighting the relative conservation of phosphosites (red triangles). The position of phosphosites identified within these proteins is labeled on top of the alignment along with the consensus sequence and normalized residue-conservation frequency at the bottom. Complete alignments including all the orthologs from higher eukaryotes (jawed vertebrates, Gnathostomata) for LC3C (n = 207) and GABARAP-L2 (n = 254) are provided in source files.

E    Sequence motif characteristic of TBK1 substrates containing the central phosphoserine position (red arrow) was computed by aligning 15-residue sequence fragments from 22 experimentally verified substrates of TBK1 from KinaseNET (a human protein kinase knowledge database: http://www.kinasenet.ca).

F    3D structures of full-length human LC3C and GABARAP-L2 highlighting sequence conservation (rainbow-colored side chains) and solvent accessibility (%) of the identified phosphosites. Conservation scores (0–9 scale) are obtained from full set of orthologous protein alignments using ConSurf [57].

Source data are available online for this figure.

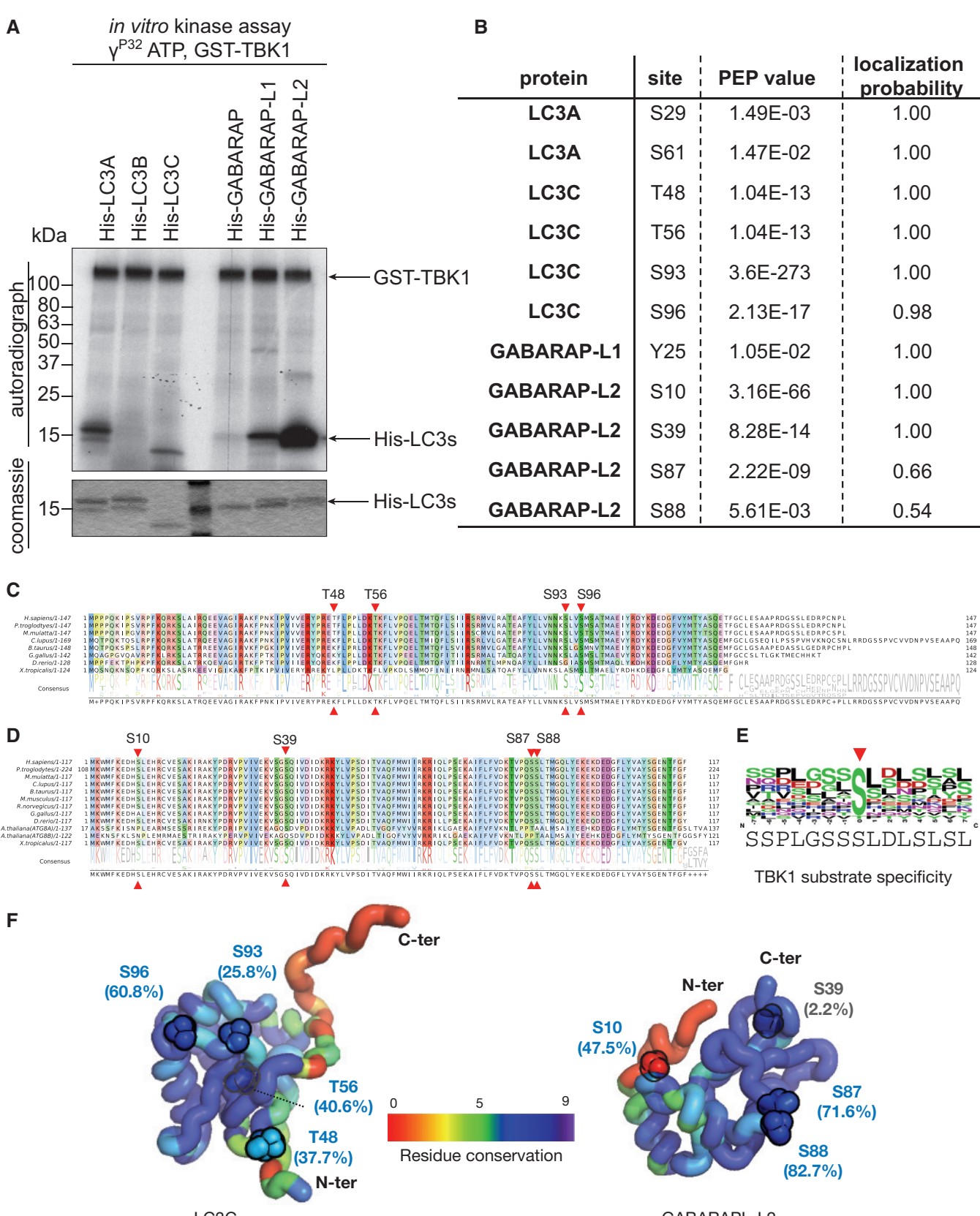

Figure 1.

**A**

| protein | site | PEP value | localization probability | TBK1 WT/ TBK1 K38A | protein ratio |
|---------|------|-----------|--------------------------|---------------------|---------------|
| **LC3C** | S93 | 1.67E-06 | 0.73 | 9.82 | 0.97 |
| **LC3C** | S96 | 1.41E-12 | 1.00 | 6.10 | 0.97 |
| **LC3C** | S98 | 1.47E-07 | 0.77 | 2.81 | 0.97 |
| **GABARAP-L2** | S10 | 8.41E-21 | 1.00 | 12.32 | 1.00 |
| **GABARAP-L2** | S39 | 2.36E-10 | 1.00 | 1.07 | 1.00 |
| **GABARAP-L2** | S87 | 6.87E-09 | 0.98 | 13.65 | 1.00 |
| **GABARAP-L2** | S88 | 5.06E-05 | 0.55 | 2.73 | 1.00 |

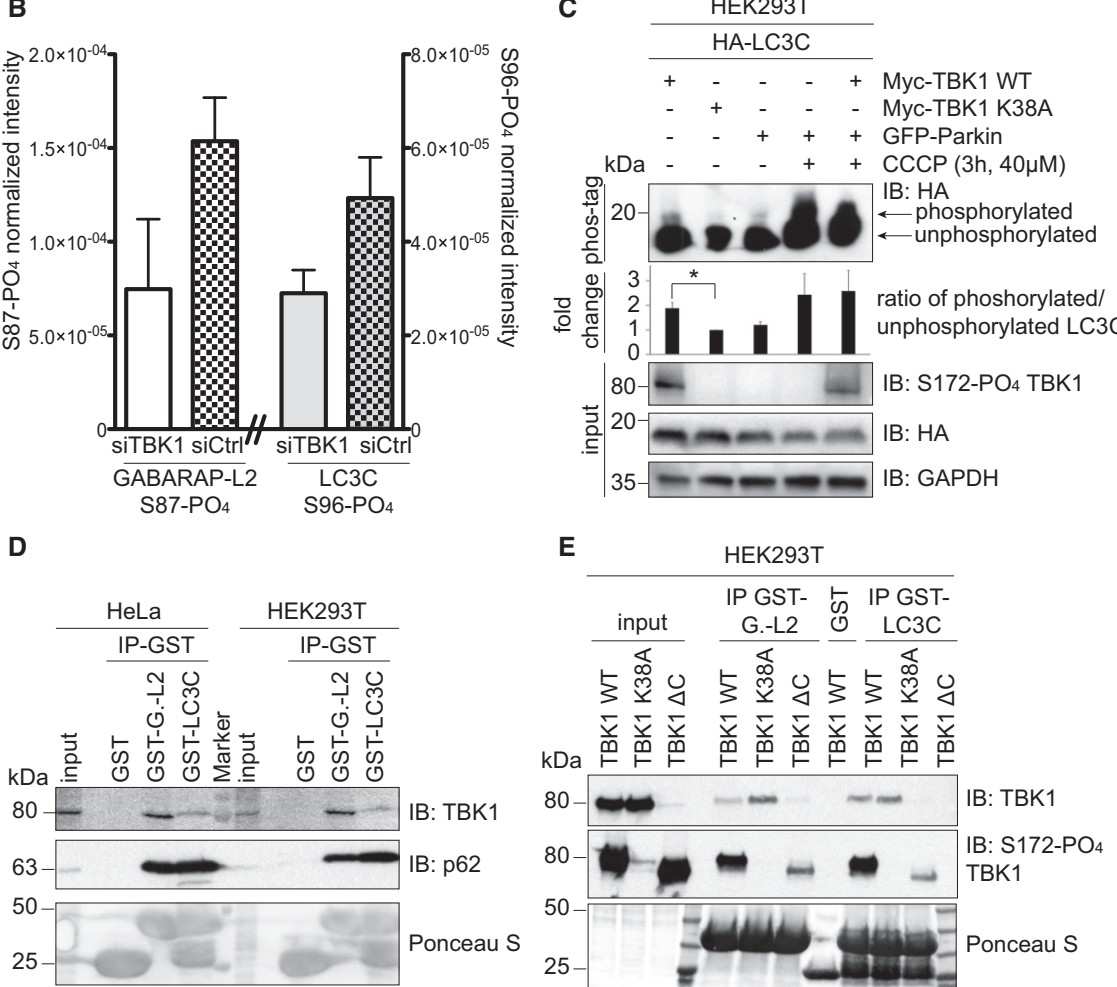

**Figure 2. TBK1 phosphorylates and binds LC3C and GABARAP-L2 *in cells*.**

A, B   Identification of phosphosites by mass spectrometry following GFP-LC3C or GFP-GABARAP-L2 immunoprecipitation. (A) TBK1 WT was overexpressed in heavy- and TBK1 kinase-dead K38A was overexpressed in light-labeled SILAC HEK293T cells. (B) HEK293T cells were treated with control or TBK1 siRNA and CCCP (3 h, 40 μM). Phosphosite intensities were normalized to total protein intensity. Data are presented as mean ± SD, $n$ = 3 biological replicates.

C   SDS–PAGE and Western blot of Phos-tag™ gel with HEK293T cell lysates. Cells were transfected with HA-LC3C, TBK1 WT or K38A, and GFP-Parkin and left untreated or treated with CCCP (3 h, 40 μM) to induce mitophagy. The ratio of phosphorylated to unphosphorylated LC3C was quantified. Data are presented as mean ± SD, $n$ = 3 biological replicates, *$P$ < 0.05, as analyzed by Student's $t$-test.

D   SDS–PAGE and Western blot of HEK293T and HeLa cell lysates and GST-LC3C or GST-GABARAP-L2 immunoprecipitations.

E   SDS–PAGE and Western blot of HEK293T cell lysates transfected with full-length TBK1, a C-terminal truncation mutant (TBK1 ΔC), a kinase-dead version (TBK1 K38A), and GST-LC3C or GST-GABARAP-L2 immunoprecipitations.

Source data are available online for this figure.

induced an upward shift and retention of phosphorylated LC3C (Fig 2C). The ratio of phosphorylated to unphosphorylated LC3C from three independent experiments was quantified and is higher upon the induction of mitophagy, by the addition of Parkin, an E3 ligase, and CCCP, a mitochondrial depolarization agent (Fig 2C). The induction of mitophagy leads to TBK1 recruitment to autophagosomes with concomitant kinase activation and results in increased LC3C phosphorylation.

Endogenous TBK1 from HeLa or HEK293T cell lysate binds to GST-LC3C and GST-GABARAP-L2 (Fig 2D) indicating direct physical interaction. The binding of TBK1 to LC3C and GABARAP-L2 is independent of its catalytic activity (Fig 2D) and could be mediated through its C-terminal coiled-coil region (Fig 2E), which is known to bind to OPTN [29].

## Phospho-mimetic LC3C impedes ATG4 cleavage and binding

To understand the consequences of phosphorylated-LC3C, we looked at its phosphosites in more detail. The LC3C phosphorylation sites S93 and S96 are situated on the face opposite to the hydrophobic pocket enabling LIR binding (Figs 1F and EV1A) and are therefore less likely to influence the direct binding of LC3C to autophagy receptors or adaptors. However, they are in close proximity to the C-terminal tail of LC3C which is proteolytically processed. ATG4-mediated processing of the LC3C C-terminal tail allows lipid conjugation and adherence to autophagosomes. To test whether phosphorylation of these residues could impair the proteolytic cleavage of the LC3C C-terminal tail by ATG4, an *in vitro* cleavage assay was performed. Double-tagged His-LC3C-Strep WT, S93/96A, or phospho-mimetic LC3C S93/96D were incubated with ATG4B for indicated times, and the C-terminal cleavage of LC3C was monitored by detecting the appearance of truncated His-LC3C protein (Fig 3A). ATG4B cleaves the entire pool of LC3C WT or S93/96A within 10 min, whereas only half of the phospho-mimetic LC3C S93/96D pool is cleaved (Fig 3A). When LC3 proteins are overexpressed in HEK293T cells, they are rapidly processed by endogenous ATG4 proteins. The C-terminal tail of LC3C that is cleaved by ATG4s is considerably larger (21 residues) than that of other LC3 family proteins. Hence, a pro-form of LC3C S93/96D can be visualized by separating the cell lysate on a 15% polyacrylamide gel (Fig 3B).

The inability of ATG4 to process phosphorylated LC3C might be distinct during stress conditions. To test this, we induced mitophagy in HEK293T cells by adding CCCP. Upon induction of mitophagy, LC3C S93/96D could not be completely processed by ATG4s (Fig 3C). Similarly, GABARAP-L2 phospho-mimetic (S88D) could not be cleaved by endogenous ATG4s, impairing subsequent lipidation (Fig 3D). We reasoned that this inability of ATG4 to process LC3C S93/96D and GABARAP-L2 S87/88D could be due to an impediment in direct protein binding, and therefore tested this by co-expressing GFP-LC3 proteins with Flag-ATG4A, Flag-ATG4B, Flag-ATG4C, and Flag-ATG4D in HEK293T cells and subjected them to GFP immunoprecipitation (Fig EV2A). Phospho-mimetic mutants LC3C S93/96D and GABARAP-L2 S87/88D displayed reduced binding to ATG4A, ATG4B, ATG4C, and ATG4D, and the $K_D$ of WT GABARAP-L2 binding to ATG4A is half the $K_D$ of GABARAP-L2 S87/88D (Fig EV2B). In addition, we performed mass spectrometry analysis of GABARAP-L2 WT, S87/88A, and S87/88D binding partners. Differential binding of GABARAP-L2 interaction partners

(significantly enriched according to a FDR 0.05 corrected Student's *t*-test with a minimal enrichment factor of 2) to S87/88D and S87/88A mutants is depicted in Fig 3E. The binding of endogenous ATG4B is significantly decreased ($P < 0.05$) in immunoprecipitated samples of GABARAP-L2 S87/88D compared to WT. Other well-known GABARAP-L2 binding partners such as TBK1, p62, NBR1, and ATG7 bind equally well to WT and S87/88D or S87/88A mutants (Fig 3E). To understand this reduced binding, we modeled the full-length LC3C-ATG4B complex based on the core crystal structure [30] (see Materials and Methods). We tested the effect of phosphorylation at both these sites (S93 and S96) by modeling phosphate groups onto serine residues in the LC3C-ATG4B complex and performed molecular dynamics (MD) simulations (up to 1.5 μs). We found that the WT LC3C-ATG4B complex with and without additional LIR interactions between ATG4B and LC3C remained stable. The C-terminal tail of LC3C remained bound and strongly anchored to the active site of ATG4B throughout the simulation. In the complex, the phosphosites S93 and S96 of LC3C (red cartoon in Fig 3F) are in close proximity to the ATG4B interacting surface (gray surface). LC3C S96 forms a hydrogen bond interaction with ATG4B E350, and LC3C S93 is close to a network of hydrogen bonds and salt bridges. In MD simulations, double phosphorylation of S93 and S96 interfered with these interactions and disrupted the binding interface between ATG4B and LC3C (Movies EV1 and EV2). The phosphorylated serine residues detached from the ATG4B surface and partially dislodged the LC3C, resulting in partial retraction of the LC3C C-terminal tail from the ATG4B active site. The negative charge introduced by phosphorylation severely weakens complex stability based on calculated binding energies (Table 1), with electrostatic interactions as the dominant factor. Figure EV2C–E shows residue-wise contributions to the binding energy mapped onto the LC3C structure. According to these calculations, phosphorylated S93 and S96 are strongly destabilizing (Fig EV2C–E; red thick cartoon), whereas unphosphorylated S93 and S96 are favorable (Fig EV2C–E; blue thin cartoon). The MD simulations and binding energy calculations indicate that phosphorylation disrupts the LC3C-ATG4B interface and destabilizes the complex.

ATG4 acts on pro-LC3s upon ribosomal release to activate and process them to LC3-I under basal conditions, as well as after autophagosome formation to recycle lipidated LC3s. However, under basal conditions TBK1 is not active to phosphorylate LC3s. TBK1 kinase activity is induced at autophagy initiation [31,32] (e.g., during mitochondrial depolarization), which is when phosphorylation potentially disrupts the LC3-ATG4 interface.

## Phosphorylation at S93 and S96 affects LC3C C-terminal tail structure and thereby impedes ATG4-mediated cleavage

Based on the simulation results for the LC3C-ATG4B complex, we hypothesized that phosphorylation of unbound LC3C could affect its C-terminal tail structure and prevent binding to the ATG4B active site. In MD simulations (see Materials and Methods) of free LC3C, we found that the C-terminal tail of LC3C (126–147) was disordered and highly dynamic (Movie EV3). By contrast, in the phosphorylated LC3C variants (S93-PO$_4$ and S96-PO$_4$), the C-terminal tail adopted more ordered conformations (Movies EV4 and EV5; Fig 4A–C). The phosphoserines formed intramolecular salt bridges with R134 (Fig 4A and B) that pulled the C-terminal tail of LC3C toward the protein, structuring it locally. In repeated simulations

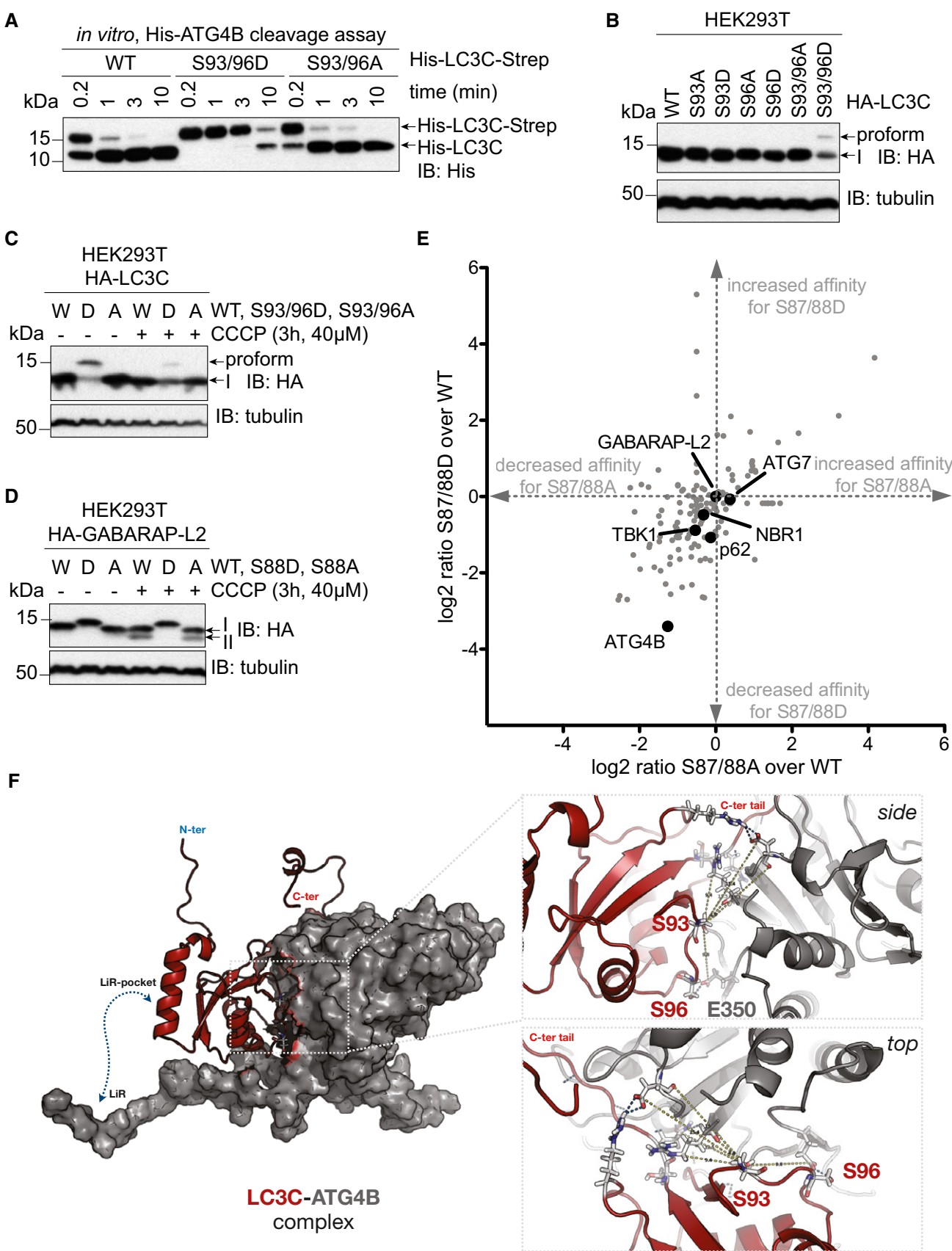

**Figure 3.**

**Figure 3.  Phospho-mimetic LC3C and GABARAP-L2 impede ATG4 cleavage and binding.**

A   SDS–PAGE and Western blot of *in vitro* ATG4 cleavage assay. Purified double-tagged His-LC3C-Strep WT and mutants were incubated with ATG4B for indicated time points. LC3C S93/96D mutation slows down C-terminal cleavage of LC3C by ATG4B.

B   SDS–PAGE and Western blot of HEK293T cell lysates transfected with LC3C WT or mutants. S93/96D mutation of LC3C impedes cleavage of pro-LC3C by endogenous ATG4s.

C, D   SDS–PAGE and Western blot of HEK293T cell lysates transfected with LC3C WT or mutants (C) or GABARAP-L2 WT or mutants (D). Cells were left untreated or treated with 40 μM CCCP for 3 h to induce mitophagy.

E   Mass spectrometry interactome analysis of HEK293T cells with overexpressed HA-GABARAP-L2 WT, S87/88A, and S87/88D. Data were analyzed with Perseus, and binding partners that specifically bind GABARAP-L2 in comparison with mock IP are depicted (*n* = 3 biological replicates).

F   Full-length LC3C (red cartoon) binds to ATG4B (gray surface) with its C-terminal tail accessible to the active site of ATG4B. Phosphorylation of LC3C at S93 and S96 (sticks) affects binding to ATG4B. Zoom-up showing the side and top view of LC3C-ATG4B interface. S96 of LC3C and E350 of ATG4B form direct hydrogen bonds across the interface. S93 position is central to a network of polar interactions (blue dashed lines; side chains shown as sticks) across the interface.

Source data are available online for this figure.

**Table 1.  Binding free energy computations for LC3C-ATG4B complexes.**

| System/Energy terms (kJ/mol) | LC3C-ATG4B (unphosphorylated) | LC3C-ATG4B (unphosphorylated + LIR) | LC3C-ATG4B (S93/S96-PO$_4$ + LIR) |
|---|---|---|---|
| van der Waals | −1,036.2 ± 110.2 | −985.9 ± 91.6 | −970.1 ± 128.4 |
| Electrostatic | −4,652.7 ± 536.4 | −4,495.8 ± 315.7 | −1,610.2 ± 293.4 |
| Polar solvation | 2,940.0 ± 406.0 | 2,883.0 ± 339.2 | 2,672.4 ± 311.0 |
| SASA | −136.2 ± 14.5 | −134.8 ± 10.2 | −125.2 ± 14.0 |
| Total binding energy | −2,885.1 ± 305.2 | −2,733.5 ± 204.0 | −33.1 ± 202.4 |
| ΔΔG | − | 151.6 ± 367.1 | 2,851.9 ± 366.2 |

The table lists different energetic contributions (mean ± SD) to the binding of LC3C and ATG4B in different phosphorylation states and with modeled LIR-WXXL interactions. The different binding energy contributions were computed using the MM-PBSA approach implemented in *g_mmpbsa* (see Materials and Methods) from MD simulations of LC3C-ATG4B complexes. The non-bonded energy terms (van der Waals and electrostatic) contribute significantly to the molecular mechanics interaction energy of the complex, whereas changes in the bonded terms (bond length, angle, and dihedral terms) do not contribute significantly to the interaction energy during complex formation.

(*n* = 6 each) of unphosphorylated and phosphorylated variants of LC3C (Fig EV3A–C), we observed a total of six salt-bridge formation events, indicating that the intramolecular salt-bridge formation between the phosphoserines and R134 is robust. We observed the salt-bridge formation on a sub-microsecond time scale (Fig 4D and E). To confirm this finding and the role of R134, we performed ATG4-mediated *in vitro* cleavage experiments of double-tagged LC3C WT, S93/96D, S93/96D R134A, and S93/96D R142A (a control mutation in the C-terminal tail). The LC3C C-terminal cleavage was monitored by the disappearance of its C-terminal Strep-tag. The mutation of S93/96D delayed the cleavage of the C-terminal tail of LC3C by ATG4B (Fig 4F). The R134A mutation could partially rescue this phenotype of S93/96D, whereas the other C-terminal tail mutation, R142A, could not (Fig 4F). The results of the ATG4-mediated cleavage assay are thus consistent with R134-phospho-serine interactions sequestering the LC3C C-terminal tail and preventing access to ATG4B and subsequent cleavage.

**Phospho-mimetic LC3C and GABARAP-L2 cannot localize to autophagosomes**

GABARAP-L2 lacks the C-terminal tail, and the ATG4B-mediated processing removes only a single C-terminal residue (F117), which exposes G116 for lipidation. Therefore, we hypothesized that phosphorylating S87 and S88 in GABARAP-L2 weakens binding to ATG4B and in turn slows down proteolytic processing. Accordingly, we tested whether phospho-mimetic GABARAP-L2 S88D and LC3C S93/96D can localize to autophagosomes, despite not being

processed by ATG4B. U2OS cells were co-transfected with HA-Parkin and GFP-GABARAP-L2 WT, S88D, S88A (Fig 5A and B) or GFP-LC3C WT, S93/96D, S93/96A (Fig 5C and D), and mitophagy was induced by the addition of CCCP for 3 h. Upon induction of mitophagy, GABARAP-L2 WT and S88A localized to autophago-somes. By contrast, the phospho-mimetic GABARAP-L2 S88D remained dispersed throughout the cell and no autophagosome formation was observed (Fig 5A and B). Likewise, phospho-mimetic LC3C S93/96D did not localize to autophagosomes upon induction of mitophagy unlike WT and S93/96A LC3C (Fig 5C and D).

**Phospho-mimetic Δ C-terminal LC3C and GABARAP-L2 are not lipidated and do not localize to autophagosomes**

Since LC3 family proteins can only be integrated into autophago-somes after C-terminal cleavage by ATG4, we tested whether artifi-cially truncated LC3C or GABARAP-L2 (Δ C-term: LC3C (1–126) and GABARAP-L2 (1–116)) could circumvent ATG4-mediated process-ing, undergo lipidation, and form autophagosomes. U2OS cells were co-transfected with HA-Parkin and GFP-GABARAP-L2 Δ C-term WT, S88A or S88D (Figs 6A and EV4A), or GFP-LC3C Δ C-term WT, S93/96A or S93/96D (Figs 6B and EV4B), and mitophagy was induced by the addition of CCCP for 3 h. Upon induction of mitophagy, GABARAP-L2 or LC3C Δ C-term WT and alanine mutants localized to autophagosomes, while phospho-mimetic mutants with truncated C-terminus (GABARAP-L2 Δ C-term S88D or LC3C Δ C-term S93/96D) remained dispersed throughout the cell and no autophagosome formation could be observed (Fig 6A and B). Upon induction of

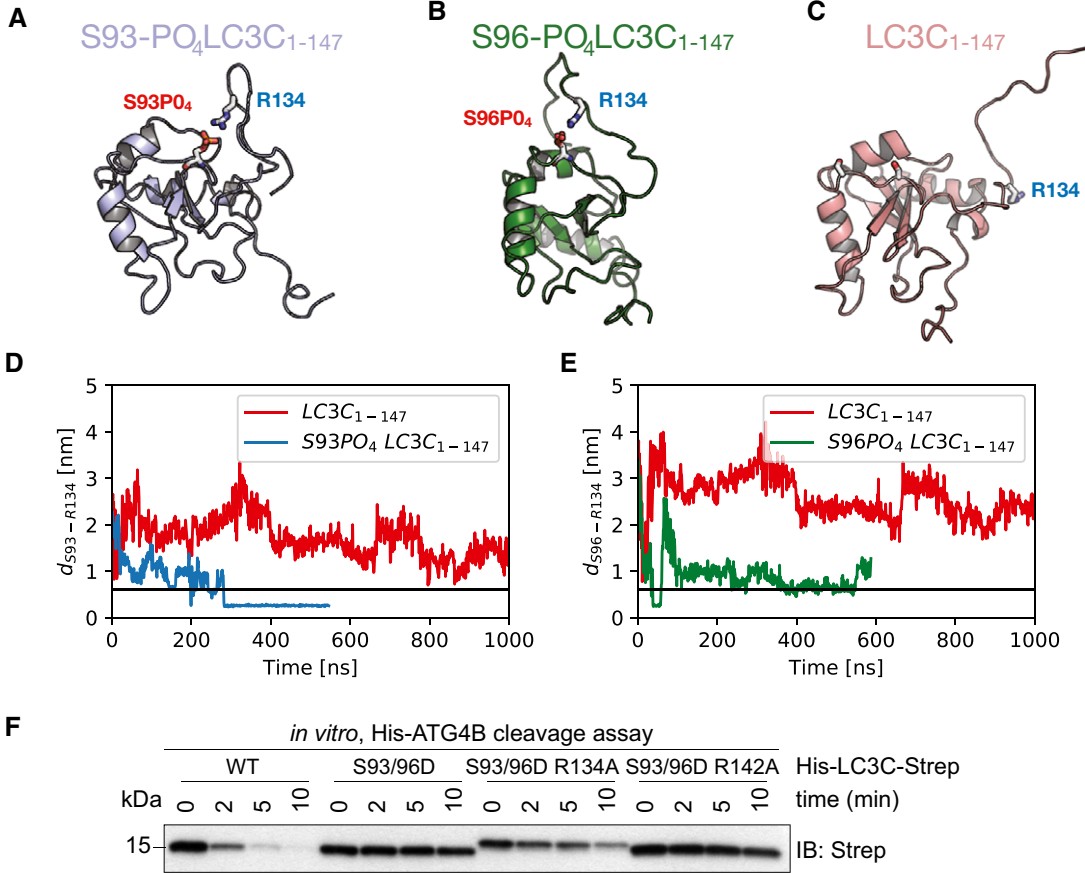

**Figure 4. Phosphorylation at S93 and S96 affects LC3C C-terminal tail structure in pro-LC3C, thereby impeding ATG4-mediated processing.**

A–C  Representative snapshots from all-atom MD simulations (see Movies EV3–EV5) of unphosphorylated (red), S93-PO$_4$ LC3C (blue), and S96-PO$_4$ LC3C (green).

D, E  Salt-bridge formation dynamics in MD simulations represented by the time-dependent minimum distance between side-chain heavy atoms of R134 to (D) S93 and (E) S96 in phosphorylated (blue) and unphosphorylated (red) LC3C simulations. The horizontal black line (0.6 nm) indicates the cut-off distance for stable electrostatic contact interactions.

F  SDS–PAGE and Western blot of *in vitro* ATG4 cleavage assay. Purified double-tagged His-LC3C-Strep WT and mutants were incubated with ATG4B in buffer for indicated time points. LC3C S93/96D R134A mutation enables C-terminal cleavage of LC3C by ATG4B.

Source data are available online for this figure.

mitophagy, GABARAP-L2 lipidation can be observed by the appearance of a lower band on Western blots (Fig 3D), which can also be observed during mitophagy induction of GABARAP-L2 Δ C-term WT and S88A, but not with phospho-mimetic GABARAP-L2 Δ C-term S88D (Fig EV4C). Hence, the phosphorylation of LC3C or GABARAP-L2 not only impedes their C-terminal cleavage by ATG4, but also their lipidation by the lipidation cascade enzymes ATG12-ATG5-ATG16L1. ATG7 function is similar to ubiquitin-activating (E1) enzymes; it recruits ATG3 (an E2-like enzyme), which then catalyzes the conjugation of the lipid moiety (PE) to the C-terminal exposed glycine of the truncated LC3 Δ C-term. Binding of the ATG12-ATG5-ATG16L1 complex (E3-like enzyme) to ATG3 enhances the lipidation of LC3, since the ATG12-ATG5-ATG16L1 complex ensures that nascently lipidated LC3 is incorporated into the phagophore membrane [4]. In order to test whether phosphorylation of LC3C/GABARAP-L2 Δ C-term impedes their processing by the lipidation cascade enzymes, we also performed an *in vitro* lipidation assay (Fig 6C). LC3C Δ C-term WT and LC3C Δ C-term

S93/96D or GABARAP-L2 Δ C-term WT and GABARAP-L2 Δ C-term S87/88D were incubated with ATP, liposomes, hATG3, hATG7, and hATG12-ATG5-ATG16L1 (reaction mix). WT LC3C and GABARAP-L2 could be successfully lipidated, while S93/96D LC3C and S87/88D GABARAP-L2 could not efficiently be conjugated to membrane *in vitro* (Fig 6C and D), indicating that phosphorylation of LC3s also affects lipid conjugation.

## TBK1-mediated GABARAP-L2 phosphorylation impedes its premature cleavage from autophagosomes by ATG4

TBK1 is recruited to the site of autophagosome formation by autophagy receptor proteins, where TBK1 phosphorylates OPTN and p62 to promote autophagy flux [19,20,22–27,31,32]. Hence, it is most likely that LC3C and GABARAP-L2 are phosphorylated by TBK1 during autophagosome formation and not during the initial processing step of pro-LC3 cleavage post-ribosomal release. In order to test whether the TBK1-mediated phosphorylation of

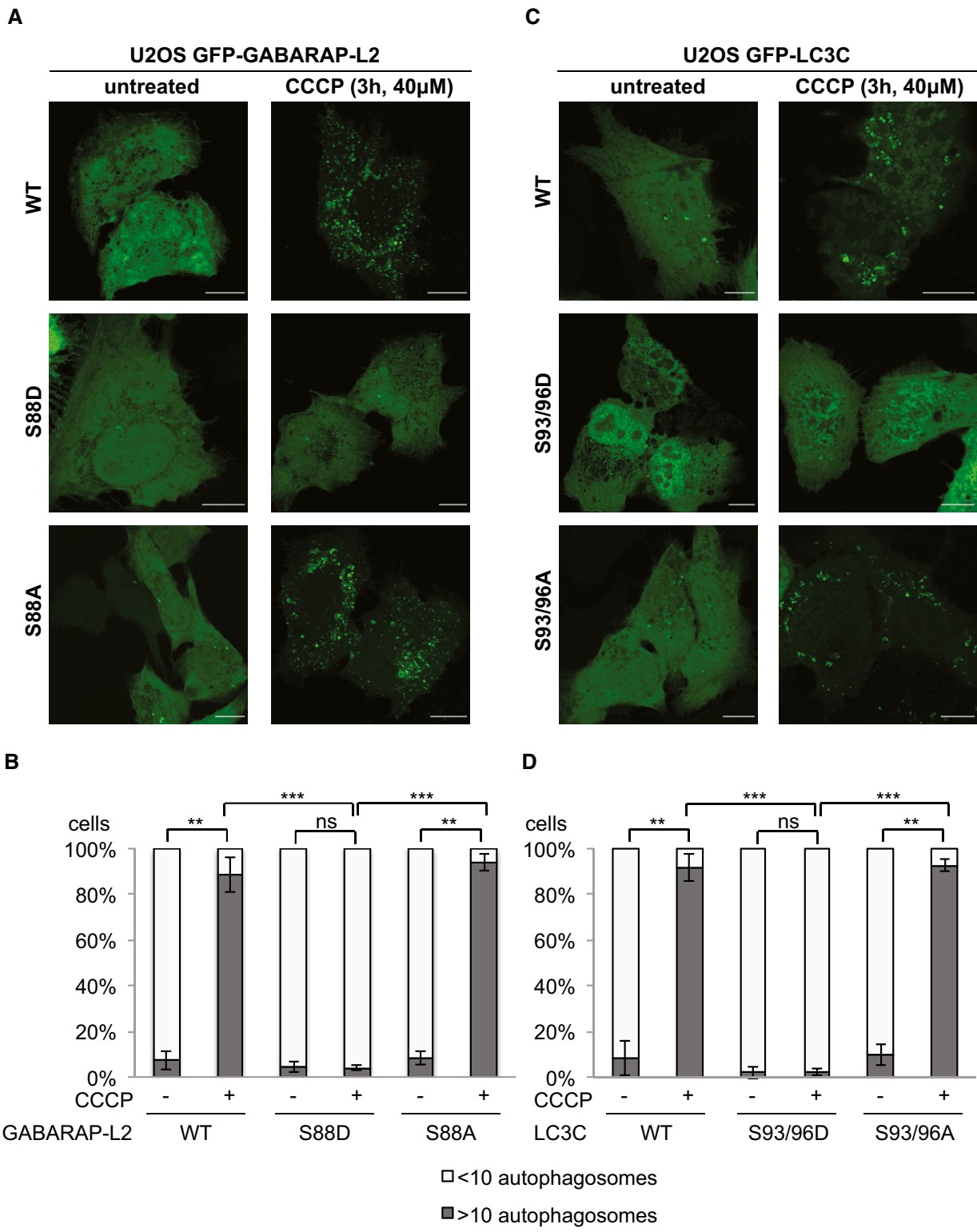

**Figure 5. Phospho-mimetic LC3C and GABARAP-L2 cannot localize to autophagosomes.**

A–D U2OS cells were transfected with GFP-GABARAP-L2 (A) or GFP-LC3C (C) WT or mutants and HA-Parkin. Mitophagy was induced by the addition of 40 μM CCCP for 3 h. WT, S87/88A GABARAP-L2 and WT, S93/96A LC3C localize to autophagosomes, whereas S87/88D GABARAP-L2 and S93/96D LC3C remain dispersed in the cytosol. Scale bar represents 10 μm. (B, D) GFP-expressing cells were counted and segregated into classes with greater and less than 10 autophagosomes per cell. Data are presented as mean ± SD, *n* = 3 biological replicates, **P < 0.01, ***P < 0.001, ns = not significant, as analyzed by Student's *t*-test.

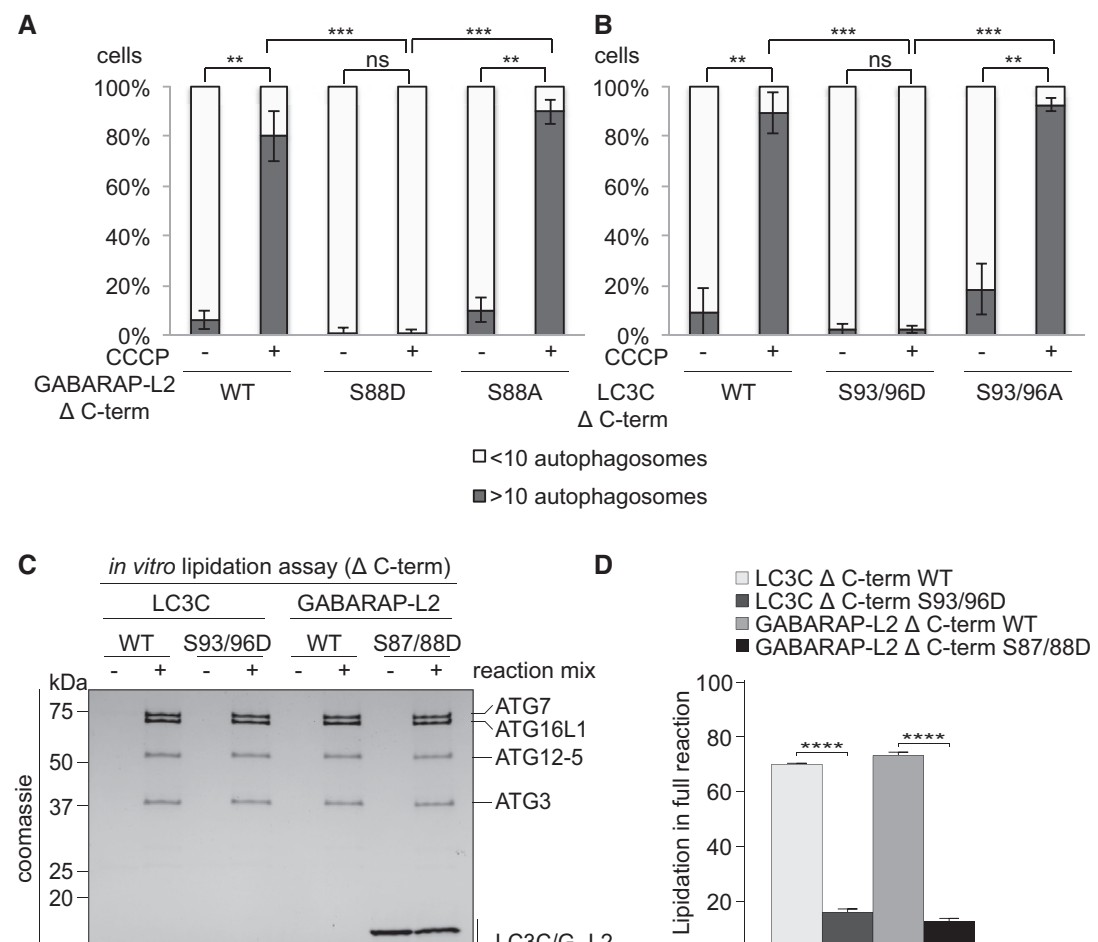

**Figure 6. Phospho-mimetic Δ C-terminal LC3C or GABARAP-L2 cannot localize to autophagosomes because they cannot be lipidated.**

A, B  U2OS cells were transfected with GFP-GABARAP-L2 Δ C-terminal (A) or GFP-LC3C Δ C-terminal (B) WT or mutants and HA-Parkin. Mitophagy was induced by the addition of 40 μM CCCP for 3 h. GFP-expressing cells were counted and segregated into classes with greater and less than 10 autophagosomes per cell. Data are presented as mean ± SD, $n$ = 3 biological replicates, **$P$ < 0.01, ***$P$ < 0.001 as analyzed by Student's $t$-test.

C  *In vitro* lipidation reactions containing 10 μM LC3C WT, LC3C S93/96D, GABARAP-L2 WT, or GABARAP-L2 S87/88D incubated with or without reaction mix (0.5 μM hATG7, 1 μM hATG3, 0.25 μM hATG12-ATG5-ATG16L1, 3 mM lipid (sonicated liposomes composed of 10 mol% bl-PI, 55 mol% DOPE, and 35 mol% POPC), 1 mM DTT, and 1 mM ATP) were incubated at 30°C for 90 min. The reactions were analyzed by SDS–PAGE and visualized by Coomassie blue stain.

D  The extent of lipidation in (C) was quantified and plotted as percentage of total protein (conjugated and unconjugated). Data are presented as mean ± SEM, $n$ = 3 biological replicates, ****$P$ < 0.0001, as analyzed by one-way ANOVA followed by Bonferroni's multiple comparison test.

Source data are available online for this figure.

---

LC3C and GABARAP-L2 has an impact on ATG4-mediated de-lipidation of LC3s from the mature autophagosome, an *in vitro* de-lipidation assay of LC3C-II and GABARAP-L2-II proteins was performed. The fractions of PE-conjugated LC3C Δ C-term WT, phospho-mimetic S93/96D and PE-conjugated GABARAP-L2 Δ C-term WT and phospho-mimetic S87/88D were enriched (see Materials and Methods) and used as substrates for the de-lipi-dating enzymes ATG4A or ATG4B (Fig 7A, D and E). WT and S87/88A GABARAP-L2-II could be de-lipidated from liposomes by ATG4B and ATG4A (at a slower rate) (Fig 7A and D) in a dose-dependent manner (Fig EV5A). On the contrary, we found that GABARAP-L2 S87/88D is not a target of ATG4A or ATG4B (Fig 7A, D and E). In addition, we tested whether TBK1 is able

to phosphorylate GABARAP-L2 and LC3C that are already incor-porated into liposomes (Fig 7B). 10 μM LC3C/GABARAP-L2-I or LC3C/GABARAP-L2-II, or 5 μM of each LC3C/GABARAP-L2-I and LC3C/GABARAP-L2-II were incubated with TBK1, and phosphory-lation was monitored by autoradiography. TBK1 phosphorylates PE-lipidated forms (-II) equally well as unlipidated (-I) LC3s (Fig 7B). In order to verify that the phospho-mimetic S87/88D GABARAP-L2-II mutant behaves similar to TBK1-phosphorylated GABARAP-L2-II, we have also tested the ability of ATG4A (Fig EV5B) and ATG4B (Figs 7C and D, and EV5B) to cleave TBK1-phosphorylated GABARAP-L2 from PE and observed that the cleavage rate is decreased in comparison with unphosphory-lated protein.

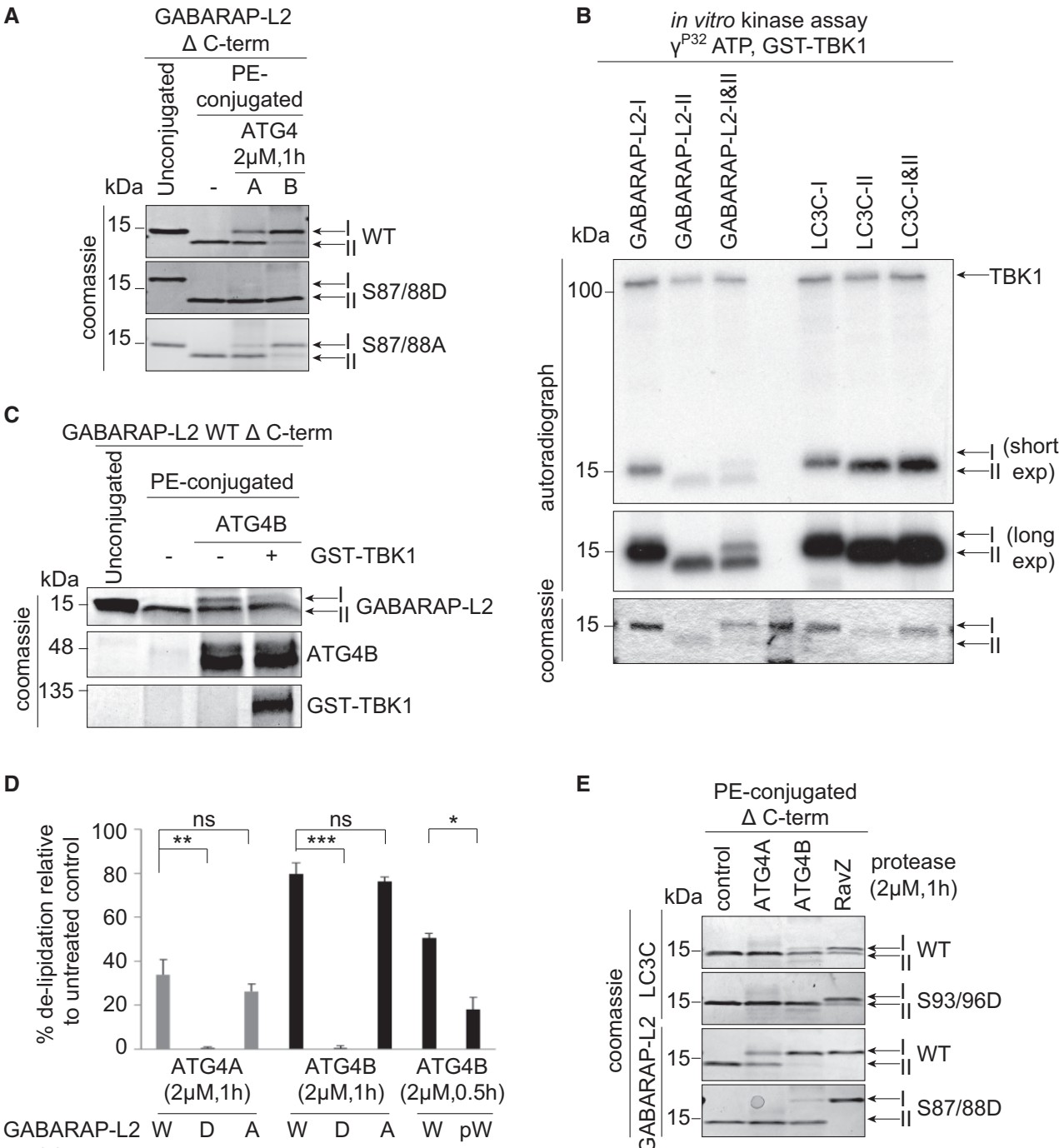

**Figure 7. TBK1-mediated GABARAP-L2 phosphorylation impedes its premature cleavage from autophagosomes by ATG4.**

A   GABARAP-L2 WT-, S87/88D-, or S87/88A-conjugated liposomes were treated or not with 2 μM ATG4A or ATG4B for 1 h at 37°C. Samples were then subjected to SDS–PAGE together with unconjugated GABARAP-L2-I.

B   Coomassie stain and autoradiography of SDS–PAGE after an *in vitro* kinase assay with GST-TBK1, PE-conjugated (II), and unconjugated (I) GABARAP-L2 and LC3C.

C   PE-conjugated GABARAP-L2-II WT was incubated in phosphorylation assay buffer in the presence or absence of GST-TBK1 for 4 h and treated or not with 2 μM ATG4B for 30 min. Samples were then subjected to SDS–PAGE together with unconjugated GABARAP-L2-I.

D   The extent of de-lipidation in (A) and (C) was quantified and plotted as percentage of total protein (conjugated and unconjugated). W = WT, D = S87/88D, A = S87/88A, pW = TBK1-phosphorylated WT GABARAP-L2. Data are presented as mean ± SD, *n* = 3 biological replicates, *P < 0.05, **P < 0.01, ***P < 0.001, ns = not significant, as analyzed by Student's *t*-test.

E   LC3C WT-, LC3C S93/96D-, GABARAP-L2 WT-, or GABARAP-L2 S87/88D-conjugated liposomes were treated or not with 2 μM ATG4A, ATG4B, or RavZ for 1 h at 37°C. Samples were then subjected to SDS–PAGE.

Source data are available online for this figure.

Finally, we also tested whether the phosphorylation of GABARAP-L2 has an impact on its de-lipidation from the phagophore by other proteases such as RavZ (Figs 7E and EV5A). RavZ is a bacterial effector protein from the intracellular pathogen *Legionella pneumophila* that interferes with autophagy by directly and irreversibly uncoupling GABARAP-L2 attached to PE on autophagosome membranes [33,34]. We found that small amounts of RavZ could remove GABARAP-L2 WT and S87/88D mutant from liposomes (Figs 7E and EV5A), indicating its effectiveness in circumventing *Legionella* growth restriction via xenophagy (when TBK1 is also activated). Likewise, RavZ is also able to cleave LC3C WT and S93/96D mutant from liposomes *in vitro* (Fig 7E), whereas ATG4B is able to de-lipidate and cleave small amounts of LC3C Δ C-term WT from liposomes, but has no activity toward LC3C Δ C-term S93/96D (Fig 7E).

### TBK1-mediated LC3 phosphorylation ensures steady autophagy flux

To assess the physiological relevance of these phosphorylation sites, we tested whether phosphorylated LC3C or GABARAP-L2 adhered to autophagosomes are still functional to perform downstream reactions. LC3 family proteins interact with autophagosome receptors such as p62, which link the growing autophagosome to cargo [35,36]. Both LC3C Δ C-term WT and phospho-mimetic S93/96D can bind to p62 (Fig 8A). Similarly, p62 and OPTN can be recruited to autophagosomes by WT as well as S87/88D GABARAP-L2 (Fig 8B) [20,37]. Once all of the cargo has been engulfed by the autophagosome, degradation can take place through the fusion with lysosomes [38]. GABARAP family proteins mediate autophagosomal–lysosomal fusion by binding to the autophagy adaptor protein PLEKHM1 [39,40]. Phosphorylation of GABARAP-L2 by TBK1 does not interfere with its ability to bind to PLEKHM1 (Fig 8B). In order to assess the physiological significance of this phosphorylation event, we have generated tandem monomeric mCherry-GFP-LC3C WT (Fig 8C) and alanine mutant cells to measure autophagic flux by FACS (Fig 8D) and high-content microscopy (Fig EV5C). Phagophores or autophagosomes that have not fused with lysosomes exhibit GFP and mCherry signal. In contrast, amphisomes or autolysosomes do not exhibit GFP fluorescence, since the GFP signal is sensitive to the acidic conditions of the lysosome lumen and only the mCherry signal is stabilized. Cells treated with bafilomycin A1 exhibit "yellow" (mCherry and GFP colocalized) puncta due to the inhibition of lysosomal acidification of the autophagosomes. LC3C S93/96A mutant cells, which cannot be phosphorylated by TBK1 upon mitophagy, exhibit fewer GFP puncta (Figs 8C and D, and EV5C). This indicates

that WT LC3C (which is phosphorylated during mitophagy) stays longer on the autophagosome and is protected from ATG4 premature de-lipidation, whereas the alanine mutant LC3C is recycled quickly and fuses with lysosomes.

Hence, TBK1-mediated phosphorylation of GABARAP-L2 and LC3C does not interfere with cargo binding but protects them from premature removal from autophagosomes by ATG4 and ensures efficient cargo engulfment with subsequent degradation.

## Discussion

The autophagy pathway is tightly regulated to ensure proper recycling and disposal of cellular material during nutrient shortage. Here, we present a new regulatory mechanism of autophagy, which influences the conjugation and de-conjugation of LC3C and GABARAP-L2 to autophagosomes. The kinase TBK1 fulfills several roles during selective autophagy. Upon autophagy induction, TBK1 is recruited to the site of autophagosome formation and gets activated by trans-autophosphorylation after accumulation [28,41]. We show that, at this stage, TBK1 can phosphorylate LC3C and GABARAP-L2 at specific serine residues to protect them from ATG4-mediated premature removal from autophagosomes.

ATG4 mediates regular processing of pro-LC3s post-ribosomal release and de-lipidation of incorrectly lipidated LC3-II on other endomembranes and favors incorporation of LC3s into autophagosomes by ATG12-ATG5-ATG16L1. Spatial and temporal regulation of recruitment and dissociation of LC3 family proteins to and from autophagosomes is achieved through regulation of ATG4 activity [13–17]. ATG4 constitutively de-conjugates LC3 family proteins from all endomembranes except from autophagosomes, to maintain a pool of unlipidated LC3 [42]. This suggests that LC3-II conjugated to autophagosomes is protected from premature de-lipidation by a timely regulatory mechanism. This regulation is achieved through the phosphorylation and dephosphorylation of ATG4 itself [15,16] and the phosphorylation of LC3s by TBK1.

The kinase activity of TBK1 is tightly regulated, and during xenophagy and mitophagy, TBK1 phosphorylates autophagy receptor proteins [19,20,22,23,25–27]. The active recruitment of TBK1 to the sites of autophagosome formation by NDP52 [19,23,31,32] hints that TBK1-mediated phosphorylation occurs on nascent phagophores, resulting in phosphorylated forms of membrane-embedded LC3s. Phosphorylation prevents premature removal of lipidated LC3C/GABARAP-L2 from growing autophagosomes by ATG4. Molecular modeling and atomistic simulations of the ATG4B-LC3C

---

**Figure 8.  TBK1-mediated LC3 phosphorylation ensures steady autophagy flux.**

A, B  SDS–PAGE and Western blot of HEK293T cell lysates and GFP IPs. (A) Cells were transfected with GFP, GFP-LC3C Δ C-terminal WT or S93/96D, and lysates used for GFP IPs. WT and S93/96D LC3C bind endogenous p62. (B) Cells were transfected with GFP, GFP-GABARAP-L2 Δ C-terminal WT or S87/88D, and lysates used for GFP IPs. WT and S87/88D GABARAP-L2 bind endogenous p62, optineurin (OPTN), and PLEKHM1.

C  U2OS cells were seeded onto coverslips and transfected with mCherry-GFP-LC3C WT or S93/96A or mCherry-GFP-GABARAP-L2 WT or S87/88A. Mitophagy was induced by the addition of 40 μM CCCP for 3 h, and lysosomal degradation of GFP was blocked by the addition of 200 nM bafilomycin A1 for 3 h. Scale bar represents 10 μm.

D  HeLa cells expressing doxycycline-inducible Parkin and stable mCherry-GFP-LC3C WT or S93/96A were treated with 1 μg/ml doxycycline and 10 μM CCCP for 16 h. mCherry and GFP fluorescence was measured by FACS, and the fold change of median fluorescence intensity of GFP over mCherry was calculated for each sample. Data are presented as mean ± SD. $n = 6$ biological replicates, **$P < 0.01$, ns = not significant, as analyzed by Student's *t*-test.

Source data are available online for this figure.

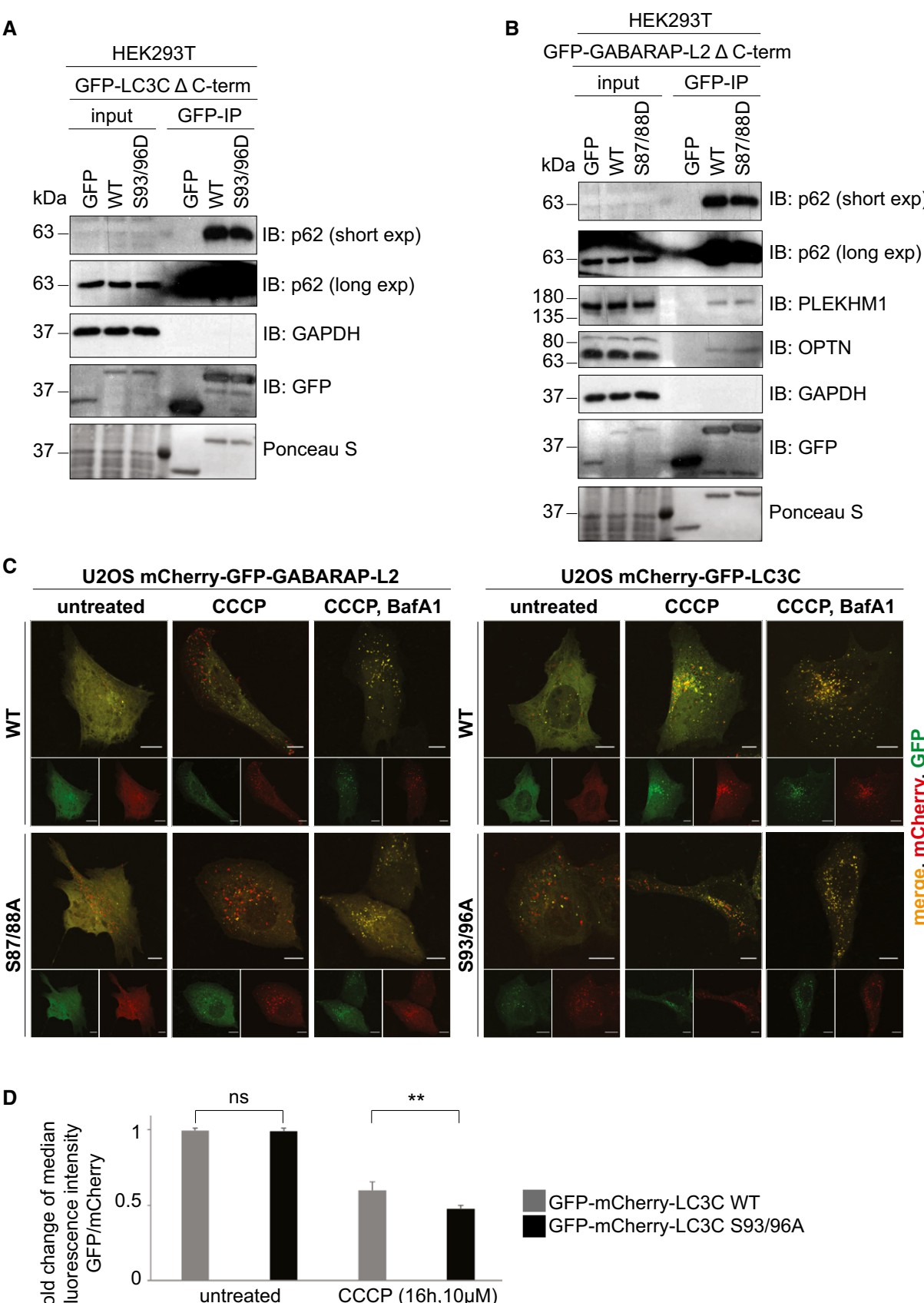

**Figure 8.**

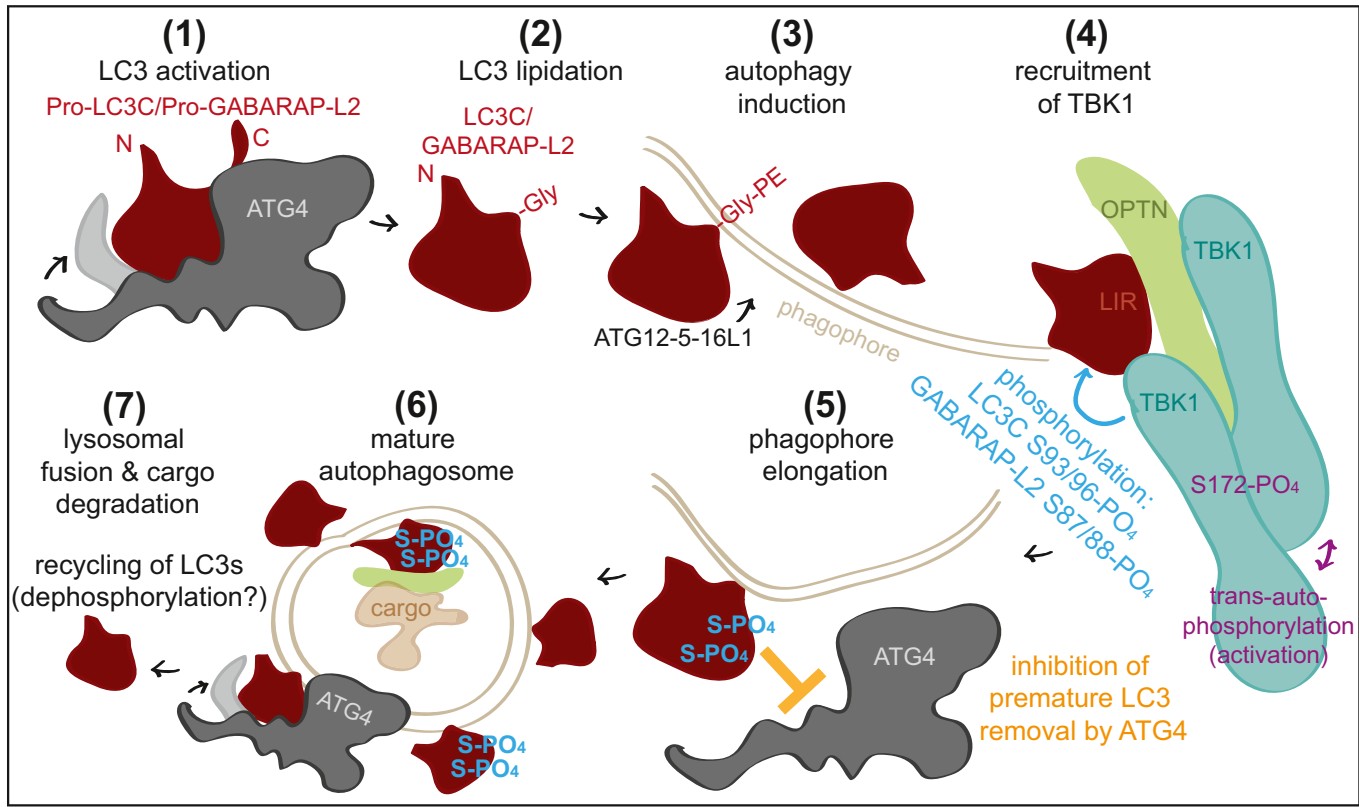

**Figure 9.  Model of TBK1-mediated LC3C and GABARAP-L2 phosphorylation.**
TBK1 recruitment to autophagosomes promotes phosphorylation of membrane-embedded LC3C and GABARAP-L2. Phosphorylation prevents premature removal of LC3C and GABARAP-L2 from autophagosomes, ensuring an unperturbed and unidirectional flow of the autophagosome to the lysosome.

complex revealed that LC3C phosphorylation impedes binding to ATG4. The weakened binding slows down de-lipidation, which ensures that a steady coat of lipidated LC3C/GABARAP-L2 is maintained throughout the early steps in autophagosome formation (Fig 9). The phosphorylation of LC3C/GABARAP-L2 does not impede their binding to autophagy receptors such as p62 or PLEKHM1 [39], which promotes unhindered downstream steps, i.e., cargo engulfment and subsequent autophagosome–lysosome fusion. Thus, phosphorylation of LC3s aids in efficient cargo engulfment and maintaining an unperturbed and unidirectional flow of the autophagosome to the lysosome.

At later stages of autophagosome formation, this process could be slowed down or reversed either by TBK1 dissociation from autophagosomes or by diminished catalytic activity. Alternatively, action of phosphatases could reverse this effect and allow de-lipidation by ATG4 prior to autophagosomal–lysosomal fusion, thereby efficiently recycling LC3 proteins.

# Materials and Methods

## Expression constructs

Expression constructs of indicated proteins were cloned into indicated vectors using PCR or the gateway system. Site-directed mutagenesis was performed by PCR to introduce desired amino acid substitutions. All expression constructs were sequenced by Seqlab.

## Protein expression and purification

GST or His-tagged fusion proteins were expressed in *Escherichia coli* strain BL21 (DE3). Bacteria were cultured in LB medium supplemented with 100 µg/ml ampicillin at 37°C in a shaking incubator (150 rpm) until OD600 $\sim$ 0.5–0.6. Protein expression was induced by the addition of 0.5 mM IPTG, and cells were incubated at 16°C for 16 h. Bacteria were harvested by centrifugation (1,800 *g*) and lysed by sonication in GST lysis buffer (20 mM Tris–HCl, pH 7.5, 10 mM EDTA, pH 8.0, 5 mM EGTA, 150 mM NaCl, 0.1% β-mercaptoethanol, 1 mM PMSF) or His lysis buffer (25 mM Tris–HCl, pH 7.5, 200 mM NaCl, 0.1% β-mercaptoethanol, 1 mM PMSF, 1 mg/ml lysozyme). For the purification of ATG4, the use of PMSF was omitted. Lysates were cleared by centrifugation (16,800 *g*), 0.05% of Triton X-100 was added, and the lysates were incubated with glutathione-Sepharose 4B beads (GE Life Sciences) or Ni-NTA agarose beads (Thermo Fisher) on a rotating platform at 4°C for 1 h. The beads were washed five times either in GST wash buffer (20 mM Tris–HCl, pH 7.5, 10 mM EDTA, pH 8.0, 150 mM NaCl, 0.5% Triton X-100, 0.1% β-mercaptoethanol, 1 mM PMSF) or in His wash buffer (25 mM Tris–HCl, pH 7.5, 200 mM NaCl, 0.05% Triton X-100, 10 mM imidazole). The immobilized proteins

were reconstituted in GST storage buffer (20 mM Tris–HCl, pH 7.5, 0.1% NaN$_3$, 0.1% β-mercaptoethanol) or eluted with His elution buffer (25 mM Tris–HCl, pH 7.5, 200 mM NaCl, 300 mM Imidazole) and dialyzed in 25 mM Tris–HCl, pH 7.5, 200 mM NaCl at 4°C overnight.

Recombinant GST-TBK1 and GST-ATG4A were obtained from the MRC PPU DSTT in Dundee, UK.

Purification of proteins used for *in vitro* lipidation/de-lipidation: Full-length hATG7, hATG3, and hATG5-12–ATG16L1 complex was expressed and purified from HEK suspension cells (HEK-F, Invitrogen) as previously described [43]. To purify mATG3, LC3C, LC3C S93/96D, GABARAP-L2 WT, GABARAP-L2 S87/88D, and RavZ, pGEX-6P-1 plasmid containing the corresponding cDNA was transformed into BL21-Gold (DE3) *E. coli*. Cells were grown at 37°C to an OD of 0.6–0.8 before induction with 0.5 mM IPTG. Cells were then grown for three additional hours before they were collected by centrifugation. Cells were resuspended in NT350 (20 mM Tris–HCl pH 7.4, 350 mM NaCl) supplemented with a Roche Complete Protease Inhibitor, lysed by sonication, and cleared by centrifugation. The supernatant was incubated at 4°C with Glutathione Beads (Sigma) for 4 h. Beads were collected and washed twice with NT350 buffer before HRV 3C protease was added and allowed to cut at 4°C overnight. The next morning, protein fractions were collected and stored at −80°C in 20% glycerol. (mATG3 and RavZ plasmids were a gift from Thomas Melia (Yale University)).

## Cell culture

HEK293T, HeLa, and U2OS cells were cultured in Dulbecco's modified Eagle's medium (DMEM; Gibco) supplemented with 10% fetal bovine serum, 2 mM L-glutamine, and 1% penicillin/streptomycin and maintained at 37°C in a humidified atmosphere with 5% CO$_2$. CCCP was resuspended in DMSO, and cells were treated with 40 μM for 3 or 16 h. Bafilomycin A1 was resuspended in DMSO, and cells were treated with 200 nM for 3 h. Plasmid transfections were performed with 3 μl GeneJuice (Merck Millipore), 0.5 μg plasmid DNA in 200 μl Opti-MEM (Life Technologies). After incubation for 15 min, the solution was added to the cells, which were lysed in lysis buffer or fixed with 4% paraformaldehyde.

## Immunofluorescence microscopy

Transfected U2OS cells were seeded onto glass coverslips in 12-well culture dishes and treated accordingly. Cells were washed in phosphate-buffered saline (PBS) before fixation with 4% paraformaldehyde for 10 min at room temperature. Cells were washed in PBS and once with deionized water before being mounted onto glass slides using ProLong Gold mounting reagent (Life Technologies), which contained the nuclear stain 4′,6-diamidino-2-phenylindole (DAPI). Slides were imaged using a Leica Microscope Confocal SP 80 fitted with a 60× oil-immersion lens.

## Cell lysis

For lysis, cells were scraped on ice in IP Lysis Buffer (50 mM HEPES, pH 7.5, 150 mM NaCl, 1 mM EDTA, 1 mM EGTA, 1% Triton X-100, 25 mM NaF, 5% glycerol, 10 μM ZnCl$_2$) or total cell lysis buffer (1% SDS, 50 mM Tris–HCl, pH 7.5, 1 mM EDTA,

25 mM NaF) supplemented with complete protease inhibitors (cOmplete, EDTA-free; Roche Diagnostics) and phosphatase inhibitors (P5726, P0044; Sigma-Aldrich). Extracts were cleared by centrifugation at 21,130 *g* for 15 min at 4°C.

## Immunoprecipitation and protein binding assays

Cleared cell extracts were mixed with glutathione-Sepharose beads (GE Healthcare) conjugated to LC3 family proteins or GST, FLAG-, or HA-agarose beads (Sigma-Aldrich) or GFP-Trap_A beads (ChromoTek) for 2 h at 4°C on a rotating platform. The beads were washed four times in lysis buffer. Immunoprecipitated and input samples were reduced in SDS sample buffer (50 mM Tris–HCl, pH 6.8, 10% glycerol, 2% SDS, 0.02% bromophenol blue, 5% β-mercaptoethanol) and heated at 95°C for 5 min [44,45].

## Western blot

For immunoblotting, proteins were resolved by SDS–PAGE and transferred to PVDF membranes. Blocking and primary antibody incubations were carried out in 5% BSA in TBS-T (150 mM NaCl, 20 mM Tris, pH 8.0, 0.1% Tween-20), and secondary antibody incubations were carried out in 5% low-fat milk in TBS-T and washings in TBS-T. Blots were developed using Western Blot Luminol Reagent (sc-2048; Santa Cruz).

## Antibodies

The following antibodies were used in this study: anti-HA-tag (11867423001; Roche), anti-FlagM2-tag (F3165; Sigma-Aldrich), anti-GFP-tag (Living Colors 632592; Clontech), anti-Strep-tag (34850; Qiagen), anti-His-tag (11922416001; Roche), anti-vinculin (V4505; Sigma), anti-tubulin (T9026; Sigma), anti-TBK1 (#3013; Cell Signaling Technology), anti-pTBK1 (pS172; #5483; Cell Signaling Technology), anti-GAPDH (#2118; Cell Signaling Technology), anti-PLEKHM1 (HPA025018; Sigma), anti-p62 (162-3; MBL), and anti-OPTN (ab23666; Abcam). Secondary HRP-conjugated antibodies goat anti-mouse (sc-2031; Santa Cruz), goat anti-rabbit (sc-2030; Santa Cruz), and goat anti-rat (sc-2006; Santa Cruz) IgGs were used for immunoblotting.

## Kinase assays

LC3-I or LC3-II family proteins were incubated in 20 μl phosphorylation buffer (50 mM Tris–HCl, pH 7.5, 10 mM MgCl$_2$, 0.1 mM EGTA, 20 mM β-glycerophosphate, 1 mM DTT, 0.1 mM Na$_3$VO$_4$, 0.1 mM ATP, or γ$^{32P}$ ATP (500 cpm/pmol)) with 1 μg of recombinant GST-TBK1 for 30 min at 30°C. The kinase assay was stopped by adding SDS sample buffer containing 1% β-mercaptoethanol and heating at 95°C for 5 min. The samples were resolved by SDS–PAGE, and the gels were stained with InstantBlue (Expedeon) and dried. The radioactivity was analyzed by autoradiography [46].

## Phosphopeptide identification

Cells were maintained in custom-made SILAC DMEM (heavy: R10, K8/light R0, K0) for 14 days, treated accordingly, and lysed

(as stated above). Incorporation of labeled amino acids to more than 95% was verified by mass spectrometry. Lysates of SILAC-labeled cells expressing GFP-tagged LC3C or GFP-tagged GABARAP-L2, TBK1 WT (heavy labeled), and TBK1 K38A (light labeled) were combined at equal amounts and incubated with GFP-Trap beads for 1 h, followed by washes under denaturing conditions (8 M Urea, 1% SDS in PBS). Bound proteins were eluted in NuPAGE LDS Sample Buffer (Life Technologies) supplemented with 1 mM DTT, boiled at 70°C for 10 min, alkylated, and loaded onto 4–20% gradient SDS–PAGE gels [19]. Alternatively, *in vitro* phosphorylated LC3 family proteins (as stated above) were used to determine TBK1-dependent phosphorylation sites. Proteins were stained using InstantBlue and digested in-gel with trypsin. Label-free phosphopeptide identification was performed with cells maintained in normal media and treated with control or TBK1 siRNA. HEK293T cells were treated with 20 nM TBK1 siRNA (sequence: 5′-GACAGAAGUUGUGAUCACATT-3′) for 24 h. Prior to cell lysis in total cell lysis buffer, cells were treated with 40 μM CCCP for 3 h and 100 nM Calyculin A (#9902; Cell Signaling Technology) for 15 min. Lysates of cells expressing GFP-tagged LC3C or GFP-tagged GABARAP-L2 were incubated with GFP-Trap beads for 1 h, followed by washes under denaturing conditions (8 M Urea). Bound proteins were reduced and alkylated with 5 mM TCEP and 20 mM chloroacetamide prior to digestion with trypsin.

SILAC, *in vitro* phosphorylated or label-free peptides were desalted on reversed-phase C18 StageTips and analyzed on an Orbitrap Elite™ or Q Exactive HF mass spectrometer (Thermo Fisher). The raw data were analyzed using MaxQuant 1.6.5.0. The database used to identify the peptides was the sequence of either GABARAP-L2 or LC3C downloaded from UniProt and a database with common laboratory contaminants. The FDR was set to 1% on protein, PSM, and site decoy level. Phosphorylation sites were validated by manual spectra interpretation.

### Interactome analysis by mass spectrometry

HEK293T cells were transfected with control plasmid, HA-GABARAP-L2 WT, HA-GABARAP-L2 S87/88A, or HA-GABARAP-L2 S87/88D plasmids, lysed in IP Lysis Buffer, and immunoprecipitated as stated above. Bound proteins were eluted in NuPAGE LDS Sample Buffer (Life Technologies) supplemented with 1 mM DTT, boiled at 70°C for 10 min, alkylated, and loaded onto 4–20% gradient SDS–PAGE gels. Proteins were stained using InstantBlue and digested in-gel with trypsin. Peptides were desalted on reversed-phase C18 StageTips and analyzed on a Q Exactive HF mass spectrometer (Thermo Fisher). The raw data were analyzed using MaxQuant 1.6.5.0 with standard settings and activated LFQ quantification. The database used to identify the peptides was the human reference protein database (UniProt downloaded December 2017), and the FDR was set to 1% on protein, PSM, and site decoy level. Statistical analysis was done with Perseus 1.6.5. Proteins were defined as interactors, if they passed a 5% FDR-corrected one-sided two-sample Student's *t*-test with a minimal enrichment factor of two compared to a HA-only IP. For interactors, differential binding of S to D/A mutants compared to WT was detected by two-sided two-sample *t*-tests.

### Phos-tag™ SDS–PAGE

Phos-tag™ acrylamide (Wako) gels were used as indicated by the supplier. Gels were prepared with 10% acrylamide, 50 μM Phos-tag™, and 100 μM MnCl$_2$. Cells were lysed in SDS sample buffer supplemented with 100 μM MnCl$_2$. Western blot lanes were quantified using ImageJ.

### Bio-layer interferometry

GST-ATG4A was obtained from the MRC PPU DSTT in Dundee, UK, and His-GABARAP-L2 WT and S87/88D proteins were purified as described above. GST-ATG4A was diluted to 1 μg/ml in bio-layer interferometry (BLI) assay buffer (300 mM NaCl, 0.004% Tween-20, 0.05% BSA in PBS) and immobilized onto anti-GST biosensors (Pall ForteBio). His-GABARAP-L2 WT and S87/88D proteins were diluted in BLI assay buffer to 20, 10, 5, and 0.625 μM concentrations in a black flat-bottom 96-well plate. The kinetic binding assay was performed on Octet RED96 instrument (Pall ForteBio). Reference wells were subtracted from sample wells, and a 1:1 global fitting model was used to determine the steady-state curve with $K_d$ values.

### ATG4 cleavage assay of unlipidated LC3C

His-LC3C-Strep double-tag proteins were purified as described above and incubated in buffer (50 mM Tris–HCl, pH 7.5, 1 mM EDTA, 150 mM NaCl, 1.5 mM DTT) at 37°C for indicated time points. The assay was stopped by adding SDS sample buffer containing 1% β-mercaptoethanol and heating at 95°C for 5 min. The samples were resolved by SDS–PAGE and imaged by Western blot.

### Modeling and MD simulations with analysis

Starting from the human LC3C (8–125) structure (PDB id: 3WAM; [47]), we added N- and C-terminal overhangs using MODELLER [48] to construct full-length LC3C (1–147). The LC3C-ATG4B complex was modeled using the core complex structure of LC3B(1–124)-ATG4B(1–357) (PDB id: 2Z0E; [30]) as template using MODELLER [48]. The C-terminal tails of both LC3C and ATG4B were modeled in extended conformations without steric clashes across their interface. Additional unresolved loops in ATG4B were modeled using the loop modeling protocol of MODELLER [48]. ATG4B contains an N- and a C-terminal LIR, both of which can, in principle, interact with the WXXL-binding sites on either non-substrate or substrate LC3. Therefore, we modeled an alternative WT complex structure including the interaction between the C-terminal LIR of ATG4B and the WXXL-binding site on substrate LC3C. Phosphoserines at positions S93 and S96 were modeled using CHARMM-GUI [49]. All structures were solvated in TIP3P water and 150 mM NaCl. After energy minimization, MD simulations of different phosphorylation states were performed using GROMACS v5.1 [50], with position-restrained NVT equilibration and NPT equilibration runs for 1,000 ps each. Production runs at 310 K and 1 atm were simulated for different times (see Table 2). We used the CHARMM36m force field [51], the Nosé–Hoover thermostat [52], the Parrinello–Rahman barostat [53], and a time step of 2 fs. For each of the LC3C systems (Table 2), six replicates were simulated with different initial velocities. We also used the molecular mechanics Poisson–Boltzmann surface area (MM-PBSA) to compute the binding

**Table 2.   Molecular dynamics simulations of LC3C and LC3C-ATG4B complexes.**

| LC3C | No of replicates | Total simulation time (µs) | Ions (Na$^+$/Cl$^-$) | TIP3P water | Total number of atoms | Salt-bridge formation events |
|---|---|---|---|---|---|---|
| WT | 6 | 7.380 | 47/51 | 17,078 | 53,723 | 0 |
| S93-PO$_4$ | 6 | 7.155 | 48/50 | 17,057 | 53,663 | 4 |
| S96-PO$_4$ | 6 | 7.420 | 48/50 | 17,101 | 53,795 | 2 |
| S93/S93D | 6 | 7.181 | 60/62 | 21,127 | 65,896 | – |
| **LC3C-ATG4B complex** | **No of replicates** | **Simulation time (µs)** | **Ions (Na+/Cl-)** | **TIP3P water** | **Total number of atoms** | **Interface interactions** |
| WT | 1 | 1.475 | 106/91 | 30,583 | 100,457 | ++ |
| WT + LIR | 1 | 1.535 | 179/164 | 56,164 | 177,357 | +++ |
| S93/S96-PO$_4$ LC3C + LIR | 1 | 1.158 | 110/89 | 31,711 | 103,863 | – |

The table lists the different simulations of LC3C$_{1-147}$ and of LC3C-ATG4B complexes, including the phosphorylation state, the number of runs, the total simulation time, the number of ions and water molecules, the total number of atoms, and the number of salt-bridge formation events between phosphorylated S93 or S96 and R134. The column "interface interactions" represents a qualitative assessment on the preservation of the interface structure and interactions in the LC3C-ATG4B complex simulations.

energies of the phosphorylated and unphosphorylated LC3C-ATG4B complexes as implemented in g_mmpbsa [54]. These binding energies contain molecular mechanical (MM), polar, and non-polar solvation energies. MM energies depend on bonded and non-bonded terms including electrostatic ($E_{elec}$) and van der Waals ($E_{vdW}$) contributions. The polar solvation energies were computed at an ionic strength of 150 mM, a solvent dielectric constant of 80, and a protein dielectric constant of 2 by solving the linearized Poisson–Boltzmann equation with a fine grid width of 0.5 Å and a coarse grid width of 1.5 times the long axis of the complex, as implemented in Assisted Poisson–Boltzmann Solver (APBS) [55]. The non-polar solvation contributions were estimated with the SASA model using a probe radius of 1.4 Å, a surface tension of $\gamma = 0.0226$ kJ/mol/Å$^2$, and an offset of 3.84 kJ/mol [56]. Binding free energies were estimated as the difference energies between bound and free states,

$\Delta G_{Binding} = G_{LC3C\text{-}ATG4B} - G_{LC3C} - G_{ATG4B}$, where the free energy contributions of the protein complex and free proteins are decomposed into a sum of molecular mechanics, solvent, and configurational entropy contributions,[§]

$$G = \Delta E_{MM} + \Delta G_{Solv} - T\Delta S$$

$$G = \Delta E_{bonded} + \Delta E_{vdW} + \Delta E_{ele} + \Delta G_{polar} + \Delta G_{non-polar} - T\Delta S$$

The binding energies were evaluated at intervals of 10 ns from the 1,000-ns MD trajectories and averaged (see Table 1). Double differences between unphosphorylated and phosphorylated complexes minimize systematic errors caused by possible energy-function inaccuracy. For the dynamic LC3C-ATG4B protein complexes studied here, these calculated free energy differences point to trends, but should not be interpreted in terms of dissociation constants.

### Liposome and proteoliposome preparation

All lipids were purchased and dissolved in chloroform from Avanti Polar Lipids (Alabaster, AL). Liposomes were prepared by

combining 55 mol% 1,2-dioleoyl-sn-glycero-3-phosphoethanolamine (DOPE), 35 mol% 1-palmitoyl-2-oleoyl-sn-glycero-3-phosphocholine (POPC), and 10 mol% bovine liver phosphoinositol (PI). The lipids were dried under nitrogen gas, and the lipid film was further dried under vacuum for 1 h. The lipids were reconstituted in NT350 buffer (350 mM NaCl, 20 mM Tris–HCl pH 7.4) and subjected to seven cycles of flash-freezing in liquid nitrogen and thawing in a 37°C bath. Liposomes were further sonicated immediately prior to the lipidation reaction.

### Lipidation reaction of LC3C and GABARAP-L2

For a full lipidation reaction, LC3C, LC3C S93/96D, GABARAP-L2 WT, or GABARAP-L2 S87/88D (10 µM) were mixed with hATG7 (0.5 µM), hATG3 (1 µM), ATG12-ATG5-ATG16L1 (0.25 µM), sonicated liposomes (3 mM), and 1 mM DTT. Lipidation was initiated by adding 1 mM ATP, and reactions were incubated at 30°C for 90 min. Samples were mixed with LDS loading buffer and immediately boiled to stop further lipidation. The reactions were run on a 4–20% SDS–PAGE gel and visualized by Coomassie blue stain and analyzed with Image Lab 6.0 (Bio-Rad).

### De-lipidation reaction of LC3C and GABARAP-L2

For a full lipidation reaction, LC3C, LC3C S93/96D, GABARAP-L2 WT, or GABARAP-L2 S87/88D (10 µM) were mixed with hATG7 (0.5 µM), mAtg3 (containing an extended N-terminal amphipathic helix that permits lipidation in the absence of ATG12-ATG5-ATG16L1) (1 µM), sonicated liposomes (3 mM), and 1 mM DTT. Lipidation was initiated by adding 1 mM ATP, and reactions were incubated at 30°C for 90 min. After the reaction was complete, the lipidation reaction was run on a Nycodenz density gradient. The bottom layer of the gradient consisted of 150 µl of 80% Nycodenz and 150 µl of the lipidation reaction. The second layer consisted of 250 µl of 30% Nycodenz, while the top layer was 50 µl of NT350 buffer. Gradients were centrifuged at 82,000 g at 4°C for 4 h in a

---

[§]Correction added on 20 November 2019, after first online publication: the equation has been corrected.

Beckman SW55 rotor. Liposomes with the conjugated LC3C or GABARAP-L2 protein were collected from the top of the tube before use in subsequent de-lipidation experiments. To measure the activity of proteases, 10 μM of proteoliposomes (concentration estimated by Coomassie blue stain) were mixed with NT350 buffer and kept on ice until activity assays were initiated by the addition of 2 μM (or indicated amounts) of either ATG4A, ATG4B, or RavZ. Reactions were incubated at 37°C for 1 h.

De-lipidation reactions of phosphorylated GABARAP-L2 were performed by adding GST-TBK1 and phosphorylation buffer to the samples for 4 h prior to the addition of 2 μM ATG4A or ATG4B for 30 min. Samples were mixed with SDS loading buffer and immediately boiled to stop proteolysis. The reactions were run on a 4–20% SDS–PAGE gel and visualized by Coomassie blue stain and analyzed with Image Lab 6.0 (Bio-Rad).

### Yokogawa CQ1 microscopy imaging

HeLa cells expressing doxycycline-inducible Parkin and stable mCherry-GFP-LC3C WT or S93/96A HeLa cells were seeded onto black, clear flat-bottom 24-well plates and treated with 1 μg/ml doxycycline and 10 μM CCCP for 16 h. Cells were incubated with Hoechst 33342 (R37605; Thermo Fisher) and washed in PBS before fixation with 4% paraformaldehyde for 15 min at room temperature. Plates were imaged using a Yokogawa CQ1 microscope. After Yokogawa CQ1 plate imaging, the *z*-stack images were analyzed using the Cell-Pathfinder software. The cell number was determined by detecting the cell nucleus stained with Hoechst, and mCherry or GFP-positive puncta were counted by the software.

### Flow cytometry (FACS)

HeLa cells expressing doxycycline-inducible Parkin and stable mCherry-GFP-LC3C WT or S93/96A were seeded onto 6-well plates ($2.6*10^5$ cells/well) and treated with 1 μg/ml doxycycline and 10 μM CCCP for 16 h. mCherry and GFP fluorescence was measured on a FACS Canto II machine and analyzed using BD FACSDiva 8.0.1 software. The fold change of median fluorescence intensity of GFP/mCherry was calculated for each sample.

### Statistical analysis

Data are presented as the mean with error bars indicating the SD (standard deviation). Statistical significance of differences between experimental groups was assessed with Student's *t*-test. Statistical analysis for *in vitro* lipidation assays was also performed by one-way ANOVA followed by Bonferroni's multiple comparison test. Differences in means were considered significant if $P < 0.05$. Differences with $P < 0.05$ are annotated as $*P < 0.01$ are annotated as $**P < 0.001$ are annotated as $***P < 0.0001$ and are annotated as $****P > 0.05$ are annotated ns (not significant). All Western blots shown are representative of biological replicates.

## Data availability

All data are available in the main manuscript text or the supplementary materials and will be made available upon request. The mass spectrometry proteomic data have been deposited to the ProteomeXchange Consortium via the PRIDE partner repository with the following dataset identifier: *in vitro* phosphorylation sites: PXD015065, www.ebi.ac.uk/pride/archive/simpleSearch?q = PXD015065; GABARAP-L2 SILAC phosphorylation sites: PXD015075, www.ebi.ac.uk/pride/archive/simpleSearch?q = PXD015075; LC3C SILAC phosphorylation sites: PXD015076, www.ebi.ac.uk/pride/archive/simpleSearch?q = PXD015076; endogenous phosphorylation sites (siTBK1 or control): PXD015077, www.ebi.ac.uk/pride/archive/simpleSearch?q = PXD015077; and GABARAP-L2 interactome data: PXD015155, www.ebi.ac.uk/pride/archive/simpleSearch?q = PXD015155.

**Expanded View** for this article is available online.

## Acknowledgements

We thank Andreas Ernst and Suchithra Guntur from Fraunhofer Institute Frankfurt for technical help with Bio-Layer Interferometry measurements, Stefan Stein and Annette Trzmiel from the FACS facility of the Georg-Speyer-Haus Frankfurt for technical help with FACS measurements, and Masato Akutsu and David McEwan for very valuable comments. This work was supported by grants from the DFG (SFB 1177 on selective autophagy), the Cluster of Excellence "Macromolecular Complexes" of the Goethe University Frankfurt (EXC 115). L.H. is supported by a European Molecular Biology Organization (EMBO) long-term postdoctoral fellowship (ALTF 1200-2014, LTFCOFUND2013, GA-2013-609409). R.M.B. and G.H. acknowledge support by the Max Planck Society and computational resources at MPCDF, Garching. A.H.L. and A.S. were supported by the Research Council of Norway (project number 221831) and through its Centers of Excellence funding scheme (project number 262652), as well as the Norwegian Cancer Society (project number 171318).

### Author contributions

LH and ID conceived the study. LH designed and performed most of the experiments. RMB developed structural models and performed MD simulations and analysis of the data with help and supervision from GH. AHL performed *in vitro* lipidation and de-lipidation assays in the laboratory of AS. UG-M performed experiments to generate Figs 2C, EV2B and EV5C. AC-P performed experiments to generate Fig 8C. FB analyzed mass spectrometry data. LH and ID wrote the manuscript with contribution from all authors. All authors approved the final version of the manuscript.

### Conflict of interest

The authors declare that they have no conflict of interest.

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
