## [Review Process File · EMBO Reports]

TBK1-mediated phosphorylation of LC3C and GABARAP-L2 controls autophagosome shedding by ATG4 protease

Lina Herhaus, Ramachandra M. Bhaskara, Alf Håkon Lystad, Uxía Gestal Mato, Adriana Covarrubias-Pinto, Florian Bonn, Anne Simonsen, Gerhard Hummer, Ivan Dikic

Review timeline:

Submission date:	18 April 2019
Editorial Decision:	20 May 2019
Revision received:	20 August 2019
Editorial Decision:	25 September 2019
Revision received:	8 October 2019
Accepted:	15 October 2019

Transaction Report:

1st Editorial Decision

20 May 2019

Thank you for the submission of your research manuscript to our journal. We have now received the full set of referee reports that is copied below.

As you will see, the referees acknowledge that the findings are potentially interesting but they all have a number of suggestions for how the study should be strengthened, most of which appear to be relatively minor and should be addressed. However, I also note that the referees indicate that the physiological significance/context of LC3C/GABARAP-L2 phosphorylation is currently unclear and I think that it would be important to strengthen this point with some further experimental evidence.

Given the constructive and supportive comments, we would thus like to invite you to revise your manuscript with the understanding that the referee concerns (as detailed above and in their reports) must be fully addressed and their suggestions taken on board. Please address all referee concerns in a complete point-by-point response. Acceptance of the manuscript will depend on a positive outcome of a second round of review. It is EMBO reports policy to allow a single round of revision only and acceptance or rejection of the manuscript will therefore depend on the completeness of your responses included in the next, final version of the manuscript.

Revised manuscripts should be submitted within three months of a request for revision; they will otherwise be treated as new submissions. Please contact us if a 3-months time frame is not sufficient for the revisions so that we can discuss the revisions further.

1) a .docx formatted version of the manuscript text (including legends for main figures, EV figures

and tables). Please make sure that the changes are highlighted to be clearly visible.

2) individual production quality figure files as .eps, .tif, .jpg (one file per figure).

4) a complete author checklist, which you can download from our author guidelines (<http://embor.embopress.org/authorguide>). Please insert information in the checklist that is also reflected in the manuscript. The completed author checklist will also be part of the RPF.

5) Please note that all corresponding authors are required to supply an ORCID ID for their name upon submission of a revised manuscript (<https://orcid.org/>). Please find instructions on how to link your ORCID ID to your account in our manuscript tracking system in our Author guidelines (<http://embor.embopress.org/authorguide>).

6) We replaced Supplementary Information with Expanded View (EV) Figures and Tables that are collapsible/expandable online. A maximum of 5 EV Figures can be typeset. EV Figures should be cited as 'Figure EV1, Figure EV2" etc... in the text and their respective legends should be included in the main text after the legends of regular figures. EV figures should be submitted as individual production quality figure files as .eps, .tif, .jpg (one file per figure, see point 2).

- For the figures that you do NOT wish to display as Expanded View figures, they should be bundled together with their legends in a single PDF file called *Appendix*, which should start with a short Table of Content. Appendix figures should be referred to in the main text as: "Appendix Figure S1, Appendix Figure S2" etc. See detailed instructions regarding expanded view here: <http://embor.embopress.org/authorguide#expandedview>.

7) We would also encourage you to include the source data for figure panels that show essential data. Numerical data should be provided as individual .xls or .csv files (including a tab describing the data). For blots or microscopy, uncropped images should be submitted (using a zip archive if multiple images need to be supplied for one panel). Additional information on source data and instruction on how to label the files are available <http://embor.embopress.org/authorguide#sourcedata>.

I look forward to seeing a revised form of your manuscript when it is ready.

REFeree REPORTS

Referee #1:

In the manuscript "TBK1-mediated phosphorylation of LC3C and GABARAP-L2 controls autophagosome shedding by ATG4 protease" Herhaus et al. study the role of LC3C and GABARAPL2 phosphorylation on their interaction and regulation by ATG4 proteases. Both, GABARAPL2 and LC3C are autophagy modifiers critical for cargo recruitment to autophagosomes and autophagosome maturation. Both proteins are processed/activated by ATG4 protease family members and at later timepoints recycled by the same set of proteins. The authors nicely show that phosphorylation in the C-terminal tail of both proteins affect ATG4-mediated processing. Phosphorylated variants appear to be not/less processed by ATG4, which indicates that phosphorylation of autophagy modifiers may support a directional flow of autophagosomes towards

lysosomes.

The paper elegantly combines *in vitro* and *in vivo* experiments to study the role of TBK1-mediated phosphorylation of LC3C and GABARAPL2 and the effects of phosphorylation sites on ATG4 activity. Whereas the majority of presented data are very convincing I would suggest minor stylistic and experimental changes prior publication.

Major points:

(1) The title is misleading. The authors claim that "autophagosome shedding" of LC3C AND GABARAPL2 is affected. However, they only show that deconjugation of GABARAPL2 is affected by phosphorylation. Thus, the authors state (starting line324): "We found that neither LC3C Δ C-term WT, nor LC3C Δ C-term S93/96D could be de-lipidated and released from the liposome by ATG4A (Figure 7A,B)." "This indicates that lipidated LC3C could be targeted specifically by other isoforms of ATG4 enzymes (A 328 TG4C or ATG4D) but not by ATG4A and ATG4B."

I would suggest to either remove LC3C from the title, or to test if other ATG4 isoforms are able to delipidate LC3C in a phosphorylation-dependent manner.

(2) Introduction (lines 91 and following): activated LC3 is commonly called LC3-I and lipidated LC3 is called LC3-II. The authors consequently do not use this nomenclature throughout the entire document. I find this more confusing than helping. Is there a specific reason why the authors did not use the established nomenclature? I would suggest to introduce it and to use it.

(3) Results (line 140 and following): PEP stands for "posterior error probability". This value operates similar to a p value. Thus, it is not a score. The smaller it is the more "secure" a hit is. The authors claim that peptides/scores were significant. Significant compared to what? Which database was used to identify these peptides, what was the FDR on peptide and site level? How sure/good can the authors localize the phosphorylation events to the mentioned sites? All this information would be more informative than the intensity of the peptides/sites and should be added.

(4) Results (line 155 and following): For me it is not clear if the mentioned phosphorylation events could also be detected *in vivo* (in cells) using endogenous protein expression levels. Data in figure 2 seem to be generated in cells overexpressing TBK1. It would be critical to show that the phosphorylation events happen *in vivo* in non-modified cells.

Minor points:

(1) Figure 2A: annotation of the table is misleading. I would suggest to explicitly state the SILAC conditions rather than the labels (H/L) and change the annotation of the last column to "protein ratio".

(2) Figure 3E: annotations of high and low exposure seem to be swapped.

(3) Line 215: the word "stabilizing" seems out of place.

Referee #2:

Summary:

Macroautophagy (often referred to as autophagy) is a conserved degradation pathway essential to the maintenance of cellular homeostasis. Autophagy degradation involves formation of a membrane sheet structure known as the phagophore, expansion of the phagophore, sequestration of cytoplasmic cargo, formation of a double membrane transport vesicle known as the autophagosome and ultimately fusion with the lysosome to enable enzymatic degradation of the cargo. The LC3 family of proteins, which is composed of 6 members in humans (LC3A, LC3B, LC3C, GABARAP, GABARAP-L1, GABARAP-L2) are small ubiquitin-like proteins that are conjugated to the lipid (phosphatidylethanoamine/PE) upon induction of autophagy. This modification enables LC3 proteins to be anchored to the phagophore and to mediate cargo selection and autophagosome formation. The ATG4 family of cysteine proteases (ATG4A, ATG4B, ATG4C, ATG4D) play crucial roles in the processing of LC3 proteins, with ATG4B being the most active. ATG4 converts pro-LC3 to active LC3 via cleaving C-terminal region of these proteins and in turn exposing a glycine residue critical to conjugation. ATG4 also delipidates LC3-PE to release LC3 from mature autophagosomes and recycle this protein. In this manuscript, Herhaus et al. showed that TBK1, a Ser/Thr protein kinase that has been shown to play a role in mediating selective autophagic degradation of mitochondria and intracellular pathogens, phosphorylates two members of the LC3 family (LC3C and GABARAP-L2). In addition to mapping the phosphorylation sites by mass spectrometry, they were able to combine structural modeling, MD simulation, and the use of phosphomimetic mutations to further understand the impact of the phosphorylation on the function

of LC3 proteins. In particular, they found that phosphorylation affects interaction between LC3C and ATG4B and that phosphomimetic mutants of LC3C and GABARAP-L2 cannot properly form autophagosomes in cells. More interestingly, even with the C-terminal pro-region removed LC3C and GABARAP-L2 phosphomimetic mutants cannot be efficiently lipidated. Lastly, they found that phosphomimetic mutations helps GABARAP-L2 persists on autophagosome (through avoiding cleavage by ATG4). The main findings are interesting and uncover a previously unrecognized function of TBK1 as well as post translational modification of the LC3 proteins. This reviewer feels that this manuscript would merit consideration for publication in EMBO Reports if the authors can address the following concerns:

Comments:

1. In the introduction, it would be helpful to provide some (even brief) information regarding what is known about the roles of the different members of the LC3/GABARAP proteins. remains unclear about the hierarchy of events that take place. This is particularly important for the audience of EMBO Reports as not all of them would be knowledgeable of the autophagy research field.
2. According to Figure 1A, LC3A, GABARP, GABARAP-L1 also phosphorylated by TBK1. It would be helpful to explain briefly the rationale to focus on LC3C and GABARAP-L2 (The PEP score was not reported for all LC3 and GABARAP proteins).
3. Are the mapped phosphorylation sites on LC3C and GABARAP-L2 conserved in other higher eukaryotes? Does TBK1 phosphorylate a consensus sequence in the solvent accessible region of LC3 proteins?
4. This reviewer might have missed it but under what conditions does TBK1 phosphorylate LC3C and GABARAP-L2? Does the level of phosphorylation go up upon induction of mitophagy?
5. The authors performed several experiments to show that the phosphomimetic mutations reduced the level of processing of pro-LC3C. While the in vitro experiments data looks convincing, the in vivo data, especially the CCP treatment data does not fully support what they described in the text. For example, there still appears to be a good level of processing of pro-LC3C according to the blot in Figure 3C. Later on in the manuscript, the authors said that they do not believe that TBK1 phosphorylates LC3 and GABARAP-L2 during the initial pro-LC3 cleavage step. Not surprisingly, the aspect of obstruction of pro-LC3 processing by TBK1-phosphorylation was not included in the model show in Figure 8. These different aspects could pose some confusion for readers.
6. The effect of the phosphomimetic mutations on LC3C/GABARAP-L2-ATG4B appears to be modest according to the pulldown data shown in Figure 3E. Would it be possible to perform biophysical experiments (eg. melting curve analysis of the LC3C-ATG4B complex) to further strengthen the data?
7. It would perhaps be helpful to move some of the cell images from the supplementary to Figure 5.
8. Figure 6E, the two forms of GABARAP-L2 run at slightly different sizes for the two reactions involving WT and S87/88 mutant. Can the authors explain why?

Referee #3:

Autophagy is a membrane traffic pathway, in which different cellular materials are sequestered within autophagosomes and then transported to lysosomes for degradation. LC3/GABARAP proteins are anchored to nascent autophagosomal membranes via their lipidation, and play multiple roles, including membrane formation, linking degradation targets to the membranes in association with autophagy receptors, and autophagosome-lysosome fusion. Lipidation of these proteins is reversible, and ATG4 is responsible for the delipidation reaction. In this study, Herhaus et al. nicely showed that two paralogs of LC3/GABARAP proteins, LC3C and GABARAP-L2, are phosphorylated by TBK1 kinase, which was known to regulate selective types of autophagy such as mitophagy and xenophagy by phosphorylating the receptor proteins p62 and Optineurin, and this

phosphorylation impedes their interactions with ATG4, blocking their delipidation by this enzyme. Taken together with previous results that TBK1 is activated at autophagosome formation sites, the authors propose the interesting, convincing model that TBK1 phosphorylates LC3C and GABARAP-L2 following their lipidation at these sites to protect them from premature delipidation.

Major comments:

- (1) It would be better if the authors can provide some evidence for the physiological significance of this phosphorylation reaction. Based on their model, if LC3C and GABARAP-L2 are not phosphorylated, they should be susceptible to premature delipidation. However, in the process where the other paralogs also work, it seems difficult to observe defects caused by premature delipidation of LC3C/ GABARAP-L2. In this light, have the authors examined if some defects in mitophagy or xenophagy could be observed in LC3C S93/96A and GABARAP-L2 S88A mutants? Or, is lysosomal transport of LC3C/GABARAP-L2 under mitophagy/xenophagy-inducing conditions decreased by the Ala mutations?
- (2) It would also be important to show that delipidation normally occurs in LC3C S93/96A and GABARAP-L2 S88A mutants *in vitro*.
- (3) The authors should examine whether in the *in vitro* system, if lipidated LC3C and GABARAP-L2 are phosphorylated by TBK1 prior to ATG4 addition, delipidation of the wild type proteins but not the Ala mutants is inhibited. This will provide direct evidence that TBK1-mediated LC3C/GABARAP-L2 phosphorylation inhibits the delipidation reaction.
- (4) This may be technically difficult, but it would be better if the authors could show that TBK1 knockdown affects phosphorylation of endogenous LC3C and GABARAP-L2.

Minor comments:

- (1) The authors clearly showed that TBK1 phosphorylates LC3C and GABARAP-L2 with high specificity among the paralogs. It would be interesting if the authors can discuss how this selectivity is determined.
- (2) Lines 142-144: The authors should clarify the position of this loop in the text, e.g., surface exposed loops connecting beta-stand X and alpha-helix Y.
- (3) In the experiments shown in Figs. 5 and 6, it is unclear how the authors count autophagosomes. If they counted puncta of GFP-LC3C/GABARAP-L2, the description that "phospho-mimetic LC3C and GABARAP-L2 cannot form autophagosomes" sounds overstated, and should be like "phospho-mimetic LC3C and GABARAP-L2 cannot localize to autophagosomes".
- (4) Line 294: "the conjugation to" should be "the conjugation of".
- (5) Line 339: "autophagosomes" should be "liposomes".
- (6) Fig. 2B: Why was HA-LC3C decreased in the third and fourth lanes?
- (7) Fig. 3E: The labels "Flag ATG4 (high)" and "Flag ATG4 (low)" should be interchanged.
- (8) Fig. 6C: The labels "S93/96" and "S87/88" should be "S93/96D" and "S87/88D".
- (9) I understand that NDP52, another autophagy receptor involved in mitophagy and xenophagy preferentially binds LC3C compared to the other paralogs. Is there any reason why the authors do not include this protein in their description/discussion?

Referee #1:

In the manuscript "TBK1-mediated phosphorylation of LC3C and GABARAP-L2 controls autophagosome shedding by ATG4 protease" Herhaus et al. study the role of LC3C and GABARAPL2 phosphorylation on their interaction and regulation by ATG4 proteases. Both, GABARAPL2 and LC3C are autophagy modifiers critical for cargo recruitment to autophagosomes and autophagosome maturation. Both proteins are processed/activated by ATG4 protease family members and at later timepoints recycled by the same set of proteins. The authors nicely show that phosphorylation in the C-terminal tail of both proteins affect ATG4-mediated processing. Phosphorylated variants appear to be not/less processed by ATG4, which indicates that phosphorylation of autophagy modifiers may support a directional flow of autophagosomes towards lysosomes.

The paper elegantly combines in vitro and in vivo experiments to study the role of TBK1-mediated phosphorylation of LC3C and GABARAPL2 and the effects of phosphorylation sites on ATG4 activity. Whereas the majority of presented data are very convincing I would suggest minor stylistic and experimental changes prior publication.

We thank the reviewer for critical reading of the manuscript.

Major points:

(1) The title is misleading. The authors claim that "autophagosome shedding" of LC3C AND GABARAPL2 is affected. However, they only show that deconjugation of GABARAPL2 is affected by phosphorylation. Thus, the authors state (starting line324): "We found that neither LC3C Δ C-term WT, nor LC3C Δ C-term S93/96D could be delipidated and released from the liposome by ATG4A (Figure 7A,B)." "This indicates that lipidated LC3C could be targeted specifically by other isoforms of ATG4 enzymes but not by ATG4A and ATG4B."

I would suggest to either remove LC3C from the title, or to test if other ATG4 isoforms are able to delipidate LC3C in a phosphorylation-dependent manner.

We agree with the referees concerns and have therefore modified our previous figure layout. In **Figure 7E** we now show that LC3C-II WT can be delipidated by ATG4B, whereas LC3C-II S93/96D cannot. Hence, we did not remove LC3C from the manuscript title, although the de-lipidation activity of ATG4B against LC3C is much lower in comparison to GABARAP-L2.

(2) Introduction (lines 91 and following): activated LC3 is commonly called LC3-I and lipidated LC3 is called LC3-II. The authors consequently do not use this nomenclature throughout the entire document. I find this more confusing than helping. Is there a specific reason why the authors did not use the established nomenclature? I would suggest to introduce it and to use it.

We have introduced the nomenclature of LC3-I/LC3-II as suggested and used it throughout the manuscript now.

(3) Results (line 140 and following): PEP stands for "posterior error probability". This value operates similar to a p value. Thus, it is not a score. The smaller it is the more "secure" a hit is.

The authors claim that peptides/scores were significant. Significant compared to what? Which database was used to identify these peptides, what was the FDR on peptide and site level? How sure/good can the authors localize the phosphorylation events to the mentioned sites? All this information would be more informative than the intensity of the peptides/sites and should be added.

We agree with the referees comment and have included the original mass spectrometry spectra for each phosphorylation site in the supplementary source files. In addition, all of the mass spectrometry raw files are now deposited on PRIDE. We have also updated **Figure 1B** and **Figure 2A** and included the relevant information. In order to export the spectra images, we re-analyzed the data using the newest Max Quant version (1.6.5.0). The information regarding database use, FDR, PSM and site decoy level has been added to the methods section.

(4) Results (line 155 and following): For me it is not clear if the mentioned

phosphorylation events could also be detected in vivo (in cells) using endogenous protein expression levels. Data in figure 2 seem to be generated in cells overexpressing TBK1. It would be critical to show that the phosphorylation events happen in vivo in non-modified cells.

In order to address this question, we have performed mass spectrometry analysis in cells treated with control or TBK1 siRNA (**Figure 2B**). TBK1 kinase activity was then induced by treatment with CCCP. pS96 of LC3C and pS87 of GABARAP-L2 were identified in cells not treated with TBK1 siRNA. We did not see pS88 of GABARAP-L2 or pS93 of LC3C, since it can be very challenging for low abundant proteins with high turnover to quantify the degree of endogenous phosphorylation (Pengo et al., 2017; Sanchez-Wandelmer et al., 2017). This also suggests frequent dephosphorylation (as it is the case for ATG4 (Pengo et al., 2017)) of LC3C or GABARAP-L2 by cellular phosphatases, which remain to be determined.

Minor points:

(1) Figure 2A: annotation of the table is misleading. I would suggest to explicitly state the SILAC conditions rather than the labels (H/L) and change the annotation of the last column to "protein ratio".

Figure 2A has been changed accordingly.

(2) Figure 3E: annotations of high and low exposure seem to be swapped.

The Figure (now EV2A) has been changed accordingly.

(3) Line 215: the word "stabilizing" seems out of place.

The text has been changed accordingly.

Referee #2:

Macroautophagy (often referred to as autophagy) is a conserved degradation pathway essential to the maintenance of cellular homeostasis. Autophagy degradation involves formation of a membrane sheet structure known as the phagophore, expansion of the phagophore, sequestration of cytoplasmic cargo, formation of a double membrane transport vesicle known as the autophagosome and ultimately fusion with the lysosome to enable enzymatic degradation of the cargo. The LC3 family of proteins, which is composed of 6 members in humans (LC3A, LC3B, LC3C, GABARAP, GABARAP-L1, GABARAP-L2) are small ubiquitin-like proteins that are conjugated to the lipid (phosphatidylethanoamine/PE) upon induction of autophagy. This modification enables LC3 proteins to be anchored to the phagophore and to mediate cargo selection and autophagosome formation. The ATG4 family of cysteine proteases (ATG4A, ATG4B, ATG4C, ATG4D) play crucial roles in the processing of LC3 proteins, with ATG4B being the most active. ATG4 converts pro-LC3 to active LC3 via cleaving C-terminal region of these proteins and in turn exposing a glycine residue critical to conjugation. ATG4 also delipidates LC3-PE to release LC3 from mature autophagosomes and recycle this protein. In this manuscript, Herhaus et al. showed that TBK1, a Ser/Thr protein kinase that has been shown to play a role in mediating selective autophagic degradation of mitochondria and intracellular pathogens, phosphorylates two members of the LC3 family (LC3C and GABARAP-L2). In addition to mapping the phosphorylation sites by mass spectrometry, they were able to combine structural modeling, MD simulation, and the use of phosphomimetic mutations to further understand the impact of the phosphorylation on the function of LC3 proteins. In particular, they found that phosphorylation affects interaction between LC3C and ATG4B and that phosphomimetic mutants of LC3C and GABARAP-L2 cannot properly form autophagosomes in cells. More interestingly, even with the C-terminal pro-region removed LC3C and GABARAP-L2 phosphomimetic mutants cannot be efficiently lipidated. Lastly, they found that phosphomimetic mutations helps GABARAP-L2 persists on autophagosome (through avoiding cleavage by ATG4). The main findings are interesting and uncover a previously unrecognized function of TBK1 as well as post translational modification of the LC3 proteins. This reviewer feels that this manuscript would merit consideration for publication in EMBO Reports if the authors can address

the following concerns:

We thank the reviewer for these comments on our manuscript and for recommending publication in EMBO Reports.

Comments:

1. In the introduction, it would be helpful to provide some (even brief) information regarding what is known about the roles of the different members of the LC3/GABARAP proteins. remains unclear about the hierarchy of events that take place. This is particularly important for the audience of EMBO Reports as not all of them would be knowledgeable of the autophagy research field.

We agree with the reviewer and have added a short paragraph (lines 92-95) to explain the different roles of the two LC3/GABARAP subgroups.

2. According to Figure 1A, LC3A, GABARAP, GABARAP-L1 also phosphorylated by TBK1. It would be helpful to explain briefly the rationale to focus on LC3C and GABARAP-L2 (The PEP score was not reported for all LC3 and GABARAP proteins). Indeed, we observe that TBK1 phosphorylates LC3A, LC3C, GABARAP-L1 and GABARAP-L2 *in vitro* and all sites are now included in the table of **Figure 1B**. Mass spectrometry analysis however revealed that GABARAP-L1 was phosphorylated on Y25, considering that TBK1 is a serine/threonine kinase, we assumed that GABARAP-L1 phosphorylation is an *in vitro* artefact. For this reason, we decided to focus our work on TBK1-mediated LC3C and GABARAP-L2 phosphorylation.

3. Are the mapped phosphorylation sites on LC3C and GABARAP-L2 conserved in other higher eukaryotes? Does TBK1 phosphorylate a consensus sequence in the solvent accessible region of LC3 proteins?

In **Figures 1C,D,E,F** we now show that the mapped phosphorylation sites on LC3C and GABARAP-L2 are conserved in other higher eukaryotes and fit to the TBK1 consensus phosphorylation motif in the solvent accessible regions of LC3C and GABARAP-L2. This information has been added to the manuscript text.

4. This reviewer might have missed it but under what conditions does TBK1 phosphorylate LC3C and GABARAP-L2? Does the level of phosphorylation go up upon induction of mitophagy?

Yes, this is true. The phosphorylation of LC3C and GABARAP-L2 is induced concomitant with the activation of TBK1, which occurs upon CCCP treatment (see **Figures 2A, 2B and 2C**). The manuscript text has been modified to explain this point with more detail.

5. The authors performed several experiments to show that the phosphomimetic mutations reduced the level of processing of pro-LC3C. While the *in vitro* experiments data looks convincing, the *in vivo* data, especially the CCP treatment data does not fully support what they described in the text. For example, there still appears to be a good level of processing of pro-LC3C according to the blot in Figure 3C. Later on in the manuscript, the authors said that they do not believe that TBK1 phosphorylates LC3 and GABARAP-L2 during the initial pro-LC3 cleavage step. Not surprisingly, the aspect of obstruction of pro-LC3 processing by TBK1-phosphorylation was not included in the model shown in Figure 8. These different aspects could pose some confusion for readers.

We agree with the referee that this could potentially be confusing for the reader. Hence, we have updated the manuscript text to further explain this (lines 308-325).

6. The effect of the phosphomimetic mutations on LC3C/GABARAP-L2-ATG4B appears to be modest according to the pulldown data shown in Figure 3E. Would it be possible to perform biophysical experiments (eg. melting curve analysis of the LC3C-ATG4B complex) to further strengthen the data?

In order to address this question, we have purified WT and phospho-mimetic GABARAP-L2 and have assessed its binding to ATG4A using Bio-Layer Interferometry (BLI) (**Figure EV2B**). Moreover, we have quantified the binding of endogenous ATG4

WT or phospho-mimetic GABARAP-L2 by mass spectrometry (**Figure 3E**). In addition, we have now included pulldown data of all 4 ATG4s to show that a phospho-mimetic mutation of LC3C or GABARAP-L2 decreases binding all ATG4s (**Figure EV2A**).

7. It would perhaps be helpful to move some of the cell images from the supplementary to Figure 5.

We have now moved the cell images to **Figure 5A, C**.

8. Figure 6E, the two forms of GABARAP-L2 run at slightly different sizes for the two reactions involving WT and S87/88 mutant. Can the authors explain why?

The introduction of an aspartic acid at position 87/88 of GABARAP-L2 results in an electrophoretic shift and changes protein mobility during SDS-PAGE. This is the case for bacterially purified and overexpressed (in mammalian cells) S87/88D GABARAP-L2.

Referee #3:

Autophagy is a membrane traffic pathway, in which different cellular materials are sequestered within autophagosomes and then transported to lysosomes for degradation. LC3/GABARAP proteins are anchored to nascent autophagosomal membranes via their lipidation, and play multiple roles, including membrane formation, linking degradation targets to the membranes in association with autophagy receptors, and autophagosome-lysosome fusion. Lipidation of these proteins is reversible, and ATG4 is responsible for the delipidation reaction. In this study, Herhaus et al. nicely showed that two paralogs of LC3/GABARAP proteins, LC3C and GABARAP-L2, are phosphorylated by TBK1 kinase, which was known to regulate selective types of autophagy such as mitophagy and xenophagy by phosphorylating the receptor proteins p62 and Optineurin, and this phosphorylation impedes their interactions with ATG4, blocking their delipidation by this enzyme. Taken together with previous results that TBK1 is activated at autophagosome formation sites, the authors propose the interesting, convincing model that TBK1 phosphorylates LC3C and GABARAP-L2 following their lipidation at these sites to protect them from premature delipidation.

We thank the reviewer for the very positive and excellent comments, which significantly improved the quality of this manuscript.

Major comments:

(1) It would be better if the authors can provide some evidence for the physiological significance of this phosphorylation reaction. Based on their model, if LC3C and GABARAP-L2 are not phosphorylated, they should be susceptible to premature delipidation. However, in the process where the other paralogs also work, it seems difficult to observe defects caused by premature delipidation of LC3C/ GABARAP-L2. In this light, have the authors examined if some defects in mitophagy or xenophagy could be observed in LC3C S93/96A and GABARAP-L2 S88A mutants? Or, is lysosomal transport of LC3C/GABARAP-L2 under mitophagy/xenophagy-inducing conditions decreased by the Ala mutations?

We have generated mCherry-GFP-LC3C WT and alanine mutant cells in order to assess the physiological significance of this phosphorylation event. We have used these cells to measure autophagic flux by FACS and high content microscopy, as now depicted in **Figures 8C,D** and **EV Figure 5C**.

(2) It would also be important to show that delipidation normally occurs in LC3C S93/96A and GABARAP-L2 S88A mutants *in vitro*.

We have now included **Figure 7A** showing *in vitro* de-lipidation of alanine mutants.

(3) The authors should examine whether in the *in vitro* system, if lipidated LC3C and GABARAP-L2 are phosphorylated by TBK1 prior to ATG4 addition, delipidation of the wild type proteins but not the Ala mutants is inhibited. This will provide direct evidence that TBK1-mediated LC3C/GABARAP-L2 phosphorylation inhibits the delipidation reaction.

We have now included **Figure 7B,C,D** and **EV Figure 5B** to show that TBK1 is able to phosphorylate lipidated LC3C and GABARAP-L2. This phosphorylation event abolishes *in vitro* de-lipidation by ATG4A and ATG4B.

(4) This may be technically difficult, but it would be better if the authors could show that TBK1 knockdown affects phosphorylation of endogenous LC3C and GABARAP-L2. This point was also raised by reviewer #1: In order to address this question, we have performed mass spectrometry analysis in cells treated with control or TBK1 siRNA (**Figure 2B**). TBK1 kinase activity was then induced by treatment with CCCP. pS96 of LC3C and pS87 of GABARAP-L2 were identified in cells not treated with TBK1 siRNA. We did not see pS88 of GABARAP-L2 or pS93 of LC3C, since it can be very challenging for low abundant proteins with high turnover to quantify the degree of endogenous phosphorylation (Pengo et al., 2017; Sanchez-Wandelmer et al., 2017). This also suggests frequent dephosphorylation (as it is the case for ATG4 (Pengo et al., 2017)) of LC3C or GABARAP-L2 by cellular phosphatases, which remain to be determined.

Minor comments:

(1) The authors clearly showed that TBK1 phosphorylates LC3C and GABARAP-L2 with high specificity among the paralogs. It would be interesting if the authors can discuss how this selectivity is determined. This is a very interesting comment raised by the reviewer. We think that this question can only be resolved by co-crystalizing TBK1 with LC3C or GABARAP-L2, as TBK1 can bind to all six family members (see Figure A below), which have a high sequence conservation (see Figure B below). Thus, although TBK1 can bind to other LC3s, the active site of the kinase might not be positioned correctly to allow phosphorylation. In addition, the other family members do not have a conserved TBK1 phosphorylation motif: GABARAP, GABARAP-L1, LC3A, and LC3B miss a leucine residue at the P+1 position. This is now explained in the manuscript.

(2) Lines 142-144: The authors should clarify the position of this loop in the text, e.g., surface exposed loops connecting beta-stand X and alpha-helix Y.

The position of the loop is in between β_3 and α_3 in both LC3C and GABARAP-L2. This has been updated in the manuscript text.

(3) In the experiments shown in Figs. 5 and 6, it is unclear how the authors count autophagosomes. If they counted puncta of GFP-LC3C/GABARAP-L2, the description that "phospho-mimetic LC3C and GABARAP-L2 cannot form autophagosomes" sounds overstated, and should be like "phospho-mimetic LC3C and GABARAP-L2 cannot localize to autophagosomes".

We agree with the referee and have changed the manuscript text accordingly.

(4) Line 294: "the conjugation to" should be "the conjugation of".

The text has been changed accordingly.

(5) Line 339: "autophagosomes" should be "liposomes".
The text has been changed accordingly.

(6) Fig. 2B: Why was HA-LC3C decreased in the third and fourth lanes?
The decrease of LC3C in these lanes occurred repeatedly, probably due to degradation or co-transfection problems. In order to achieve even levels of LC3C we transfected cells with HA-LC3C 48 hours prior to lysis and only added GFP-Parkin and/or Myc-TBK1 for 24 hours prior to lysis. Thereby, even LC3C expression is assured and the Figure has been updated with new **Figure 2C**. In addition, we have quantified the ratio of phosphorylated to unphosphorylated LC3C from 3 experiments and added this information to the figure.

(7) Fig. 3E: The labels "Flag ATG4 (high)" and "Flag ATG4 (low)" should be interchanged.

Figure 3E (now EV2A) has been changed accordingly.

(8) Fig. 6C: The labels "S93/96" and "S87/88" should be "S93/96D" and "S87/88D".
Figure 6C has been changed accordingly.

(9) I understand that NDP52, another autophagy receptor involved in mitophagy and xenophagy preferentially binds LC3C compared to the other paralogs. Is there any reason why the authors do not include this protein in their description/discussion?
There was no specific reason why we had not mentioned NDP52 and we have now included this in the manuscript discussion.

2nd Editorial Decision

25 September 2019

Thank you for the submission of your revised manuscript to EMBO reports. We have now received the full set of referee reports that is copied below. Since reviewer 2 was unfortunately not available anymore, I have asked referee 1 and 3 to also take your response to this referee into account.

As you will see, both referees are very positive about the study and request only minor changes to clarify text and figures.

From the editorial side, there are also a few things that we need before we can proceed with the official acceptance of your study.

- Figure callouts: We noticed that Fig 7C is never mentioned in the text.

- Movies: Please remove the movie legends from the manuscript and provide them as simple README.txt file. Then zip movie and legend together and upload the .ZIP files. Please also note the nomenclature "Movie EV1" etc.

- Please update the references to the numbered format of EMBO reports. The abbreviation 'et al' should be used for more than 10 authors. You can download the respective EndNote file from our Guide to Authors

<https://drive.google.com/file/d/0BxFM9n2IEE5oOHM4d2xEbmpxN2c/view>

- Thank you for providing all source data; this is much appreciated. However, please note that we require one source data file per figure, i.e., all source data for Figure 1 should be in one file etc.

- I also noticed that the source data for Figure 2 is incorrectly assigned (2B should be 2C etc.)

- Please provide URLs that resolve to the respective datasets in the Data Availability section.

- Our data editors from Wiley have already inspected the Figure legends for completeness and accuracy. Please see their suggested changes in the attached Word file. Please note that the attached Word file represents an earlier version of the document. Before the manuscript was sent back to the

reviewers you made some changes (data availability section; the number of experiments in EV5C was changed to 3; Florian Bonn was added to the manuscript title page).

- Finally, EMBO reports papers are accompanied online by A) a short (1-2 sentences) summary of the findings and their significance, B) 2-3 bullet points highlighting key results and C) a synopsis image that is 550x200-400 pixels large (width x height). You can either show a model or key data in the synopsis image. Please note that the size is rather small and that text needs to be readable at the final size. Please send us this information along with the revised manuscript.

REFEREE REPORTS

Referee #1:

My comments were adequately addressed. I have only two minor remarks which should be addressed prior acceptance:

- (1) Newly presented Figure 3E: data should be normalized to GABARAPL2, only taking common peptides between the mutated variants into account.
- (2) Page 7, line 189-198: repetition of text should be corrected.

Referee #3:

The authors have addressed the issues I raised for the original manuscript in a satisfactory manner. Being requested by Martina, I also assessed the authors' response to the comments from referee 2, and found that the authors have adequately addressed his/her comments, too.

2nd Revision - authors' response

8 October 2019

The authors performed all minor editorial changes.

Corresponding Author Name: Ivan Dikic

Manuscript Number: EMBOR-2019-48317V1